# ETO family protein Mtgr1 mediates Prdm14 functions in stem cell maintenance and primordial germ cell formation

**Nataliya Nady[1†], Ankit Gupta[2†], Ziyang Ma[1], Tomek Swigut[1], Akiko Koide[2], Shohei Koide[2]\*, Joanna Wysocka[1,7,3,4]\***

[1]Department of Chemical and Systems Biology, Stanford University School of Medicine, Stanford, United States; [2]Department of Biochemistry and Molecular Biology, University of Chicago, Chicago, United States; [3]Department of Developmental Biology, Stanford University School of Medicine, Stanford, United States; [4]Institute of Stem Cell Biology and Regenerative Medicine, Stanford University School of Medicine, Stanford, United States; [7]Howard Hughes Medical Institute, Chevy Chase, United States

**Abstract** Prdm14 is a sequence-specific transcriptional regulator of embryonic stem cell (ESC) pluripotency and primordial germ cell (PGC) formation. It exerts its function, at least in part, through repressing genes associated with epigenetic modification and cell differentiation. Here, we show that this repressive function is mediated through an ETO-family co-repressor Mtgr1, which tightly binds to the pre-SET/SET domains of Prdm14 and co-occupies its genomic targets in mouse ESCs. We generated two monobodies, synthetic binding proteins, targeting the Prdm14 SET domain and demonstrate their utility, respectively, in facilitating crystallization and structure determination of the Prdm14-Mtgr1 complex, or as genetically encoded inhibitor of the Prdm14-Mtgr1 interaction. Structure-guided point mutants and the monobody abrogated the Prdm14-Mtgr1 association and disrupted Prdm14's function in mESC gene expression and PGC formation in vitro. Altogether, our work uncovers the molecular mechanism underlying Prdm14-mediated repression and provides renewable reagents for studying and controlling Prdm14 functions.

**\*For correspondence:** skoide@ uchicago.edu (SK); wysocka@ stanford.edu (JW)

[†]These authors contributed equally to this work

## Introduction

Prdm14 is a sequence-specific transcriptional regulator that plays key roles in promoting primordial germ cell (PGC) specification and safeguarding pluripotency of embryonic stem cells (ESCs) (*Nakaki and Saitou, 2014*). During mouse embryogenesis, Prdm14 is expressed in preimplantation embryos, where its asymmetric expression promotes allocation of cells toward the pluripotent inner cell mass (ICM) fate (*Burton et al., 2013*; *Nakaki and Saitou, 2014*). Prdm14 expression ceases in postimplantation epiblast cells and their differentiated progeny. However, during PGC specification from the epiblast, cells reacquire many transcriptional and epigenetic characteristics of the preimplantation state, and Prdm14 is re-expressed along with several other pluripotency-associated factors (reviewed in [*Magnúsdóttir and Surani, 2014*; *Saitou et al., 2012*]). The loss of Prdm14 in mice results in sterility associated with early germ cell deficiency, as cells fated to become PGCs fail to reacquire expression of key pluripotency factors and undergo epigenetic reprogramming (*Yamaji et al., 2008*). Furthermore, overexpression of Prdm14 in epiblast-like cells (EpiLCs) is sufficient to induce PGCs in vitro (albeit with low frequency), suggesting a central role of Prdm14 in the

**eLife digest** In animals, there are many different types of cells that perform different roles. For example, stem cells divide to produce new cells that may then become other types of cells such as muscle or skin cells. Most stem cells can only produce a limited range of other cell types, except for a subset known as 'pluripotent' stem cells that can give rise to cells of any type in the body.

A protein called Prdm14 helps to keep stem cells in a pluripotent state. In mouse embryos, Prdm14 binds to and represses particular genes that promote a process by which the stem cells can change into other cell types. If Prdm14 is missing from pluripotent stem cells, these cells become more sensitive to signals that encourage them to become other types of cells, which can lead to the loss of pluripotency. Prdm14 contains a region called the SET domain. In other proteins, this domain can alter how DNA is packaged to help switch particular genes on or off. However, such activity has not been found for the SET domain of Prdm14, raising questions about how it actually works.

Here, Nady, Gupta et al. show that Prdm14 tightly interacts with a protein called Mtgr1, which belongs to a family of proteins known to be involved in leukemia. The loss of Mtgr1 also leads to the loss of pluripotency in mouse stem cells and disrupts the formation of reproductive stem cells. Furthermore, Mtgr1 binds to the same genes as Prdm14. Next, Nady, Gupta et al. made synthetic proteins, termed monobodies, that bind to the Prdm14 SET domain. One such monobody enabled the authors to determine the three-dimensional structure of Prdm1 and Mtgr1, which revealed that the SET domain of Prdm14 has many points of contact with Mtgr1. Importantly, interaction between the two partners is crucial for these proteins to maintain pluripotency and promote the production of reproductive stem cells.

Altogether, these findings identify Mtgr1 as a key binding partner of Prdm14 in pluripotent stem cells and uncover a role for the SET domain in this interaction. A future challenge will be to understand the roles of these proteins in leukemia and other diseases.

mouse PGC regulatory network (*Magnúsdóttir et al., 2013*; *Nakaki et al., 2013*). Furthermore, Prdm14 is repressed in normal somatic tissues but is aberrantly reactivated in human malignancies of various tissue origin, including leukemias and lymphomas, breast, testicular, and lung cancers (*Carofino et al., 2013*; *Dettman et al., 2011*; *Nishikawa et al., 2007*; *Ruark et al., 2013*; *Zhang et al., 2013*).

Given the poor accessibility and transient nature of preimplantation embryo cells and PGCs in vivo, mechanistic studies of Prdm14 function in early development have been chiefly conducted in the context of mouse ESCs (mESCs). These cells represent a so-called 'naïve' pluripotent state, thought to resemble preimplantation embryo ICM and serve as a useful system for understanding early cell fate decisions (*Nichols and Smith, 2009*). Loss of Prdm14 destabilizes mESCs and sensitizes them to differentiation stimuli, leading to acquisition of alternative embryonic states, such as the postimplantation epiblast state or extraembryonic endoderm state, and eventual depletion of the naïve cell subpopulation (*Ma et al., 2011*; *Yamaji et al., 2013*). The differentiation in Prdm14$^{-/-}$ cells is thought to result from upregulation of signaling pathways such as the fibroblast growth factor receptor (FGFR) pathway and by widespread DNA hypermethylation (*Grabole et al., 2013*; *Hackett et al., 2013*; *Leitch et al., 2013*). Indeed, genome-wide Prdm14 occupancy studies by chromatin immunoprecipitation with sequencing (ChIP-seq) suggest that these are direct effects of the loss of Prdm14, as Prdm14 occupies and represses the regulatory elements of genes involved in FGFR signaling and de novo DNA methylation (*Leitch et al., 2013*; *Ma et al., 2011*; *Magnúsdóttir et al., 2013*; *Yamaji et al., 2013*). Nonetheless, Prdm14$^{-/-}$ mESCs can be maintained indefinitely under 2i conditions (*Grabole et al., 2013*; *Payer et al., 2013*; *Yamaji et al., 2013*), in which differentiation stimuli, including FGFR signaling, are chemically inhibited, providing an opportunity to study the early effects of Prdm14 deficiency upon release from such inhibition.

Although the cellular and molecular phenotypes associated with loss of Prdm14 in mESCs have been well characterized, much less is known about molecular mechanisms and partners through which Prdm14 acts. As a member of the PRDM family, Prdm14 contains both a zinc-finger array, responsible for sequence-dependent DNA binding (*Ma et al., 2011*), and a PR domain that is

related to the SET domain (Su(var)3–9, Enhancer-of-zeste and Trithorax) (*Fog et al., 2012*; *Hohenauer and Moore, 2012*). Many SET domains harbor methyltransferase activity for either histone or non-histone substrates (*Del Rizzo and Trievel, 2011*). However, to date, no enzymatic activity has been reported for Prdm14, and interestingly multiple members of the PRDM family appear to be catalytically inactive. Instead, candidate-based co-immunoprecipitation studies implicated Polycomb complex PRC2 as a mediator of Prdm14-dependent repression (*Chan et al., 2013*; *Yamaji et al., 2013*). Nonetheless, it remains unclear whether PRC2 is a major or auxiliary partner of Prdm14, and what other molecular players are important for Prdm14's function.

To address these questions, we used an unbiased biochemical approach to uncover major Prdm14-associated proteins in mESCs. We identified an ETO-family corepressor, myeloid translocation gene related 1 (Mtgr1, a.k.a. Mtg8r, Cbfa2t2, and Zmynd3), as a direct, stoichiometric partner of Prdm14. We demonstrate that Mtgr1 co-occupies Prdm14 target loci, and its deletion in mESCs results in phenotypes and gene expression defects similar to those observed upon loss of Prdm14. Moreover, Mtgr1 knockout cells show impaired induction of PGC-like cells in vitro. To further facilitate studies of the Prdm14-Mtgr1 complex, we mapped interaction domains and developed multiple synthetic binding proteins, termed monobodies, that specifically recognize the SET domain of Prdm14 in a manner independent of, or alternatively, competitive with Mtgr1. Taking advantage of the stabilizing effect of one such monobody, we obtained a crystal structure of the Prdm14-Mtgr1 complex, revealing an extensive interface and electrostatic interactions mediating the association of the two proteins. Furthermore, structure-guided mutagenesis of the interface and the use of an inhibitory monobody demonstrated the function of the complex in safeguarding pluripotency and PGC-like cell induction. Altogether, we report a multi-disciplinary study that advances our understanding of Prdm14 function, identifying Mtgr1 as the major partner of Prdm14 in its roles in pluripotency and PGC induction, and providing the community with renewable, genetically encoded reagents that can be used both in vitro and in vivo to study and control Prdm14 function in development and malignancy.

## Results

### Identification of Mtgr1 as a novel Prdm14 partner

To identify Prdm14 partners in an unbiased manner, we employed a two-step immunopurification strategy (FLAG followed by HA [FH]) from a previously described clonal mESC transgenic line stably expressing tagged FH-Prdm14 (*Ma et al., 2011*). Examination of recovered proteins by sodium dodecyl sulfate polyacrylamide gel electrophoresis (SDS-PAGE) and silver staining revealed two major polypeptides that were present in similar quantities in the FH-Prdm14 purifications, but not in the control immunoprecipitates (*Figure 1A*). These polypeptides were subsequently identified by mass spectrometry as Prdm14 and Mtgr1 (*Figure 1A*, *Figure 1—source data 1*), the latter of which is one of the three members of the ETO family of co-repressors (*Davis et al., 2003*). Mass spectrometry analysis also identified additional polypeptides enriched uniquely in the FH-Prdm14 purifications, including the other two ETO proteins Mtg8 and Mtg16, their known repressive complex partners Tbl1/Tblr1 and histone deacetylases (HDACs), as well as Brg1 complex components and Oct4, among others (*Figure 1—source data 1*). Of note, we did not detect components of the Polycomb complex PRC2 (*Chan et al., 2013*; *Yamaji et al., 2013*).

Mtgr1 was the major polypeptide identified in our analysis and its stoichiometric recovery in our purifications indicated it might represent a strong and direct partner of Prdm14. To confirm this, we first verified Prdm14 and Mtgr1 association using reciprocal co-immunoprecipitations from mESC nuclear extracts (*Figure 1B*). Next, we mapped the minimal regions within Prdm14 and Mtgr1 that were required for the interaction by overexpressing differentially-tagged proteins (V5-Mtgr1 and FH-Prdm14) in HEK293 cells, followed by IP-Western analysis (*Figure 1C, D*). This strategy revealed that the nervy homology region 1 (NHR1) domain of Mtgr1 was necessary and sufficient for the interaction with Prdm14, whereas both Prdm14 SET domain and the region directly preceding it (pre-SET) were important for efficient binding to Mtgr1 (*Figure 1C, D*). To quantify the strength of the Prdm14–Mtgr1 interaction, we next expressed and purified recombinant proteins corresponding to the NHR1 domain of Mtgr1 (residues 98–206) and pre-SET+SET domains of Prdm14 (residues 184–373), and performed a bead-based binding assay in reciprocal orientations (*Nishikori et al., 2012*).

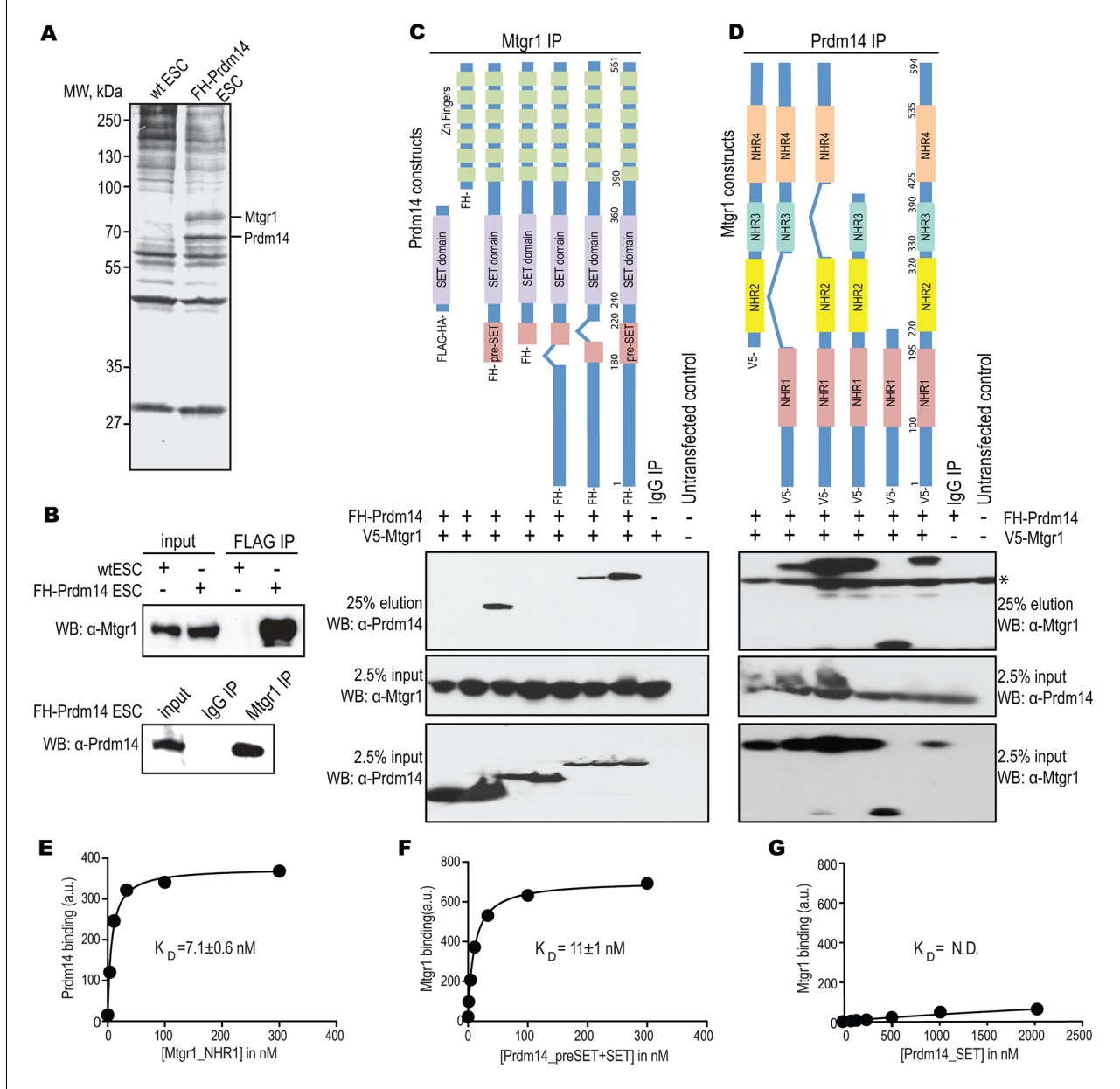

**Figure 1.** Prdm14 directly binds to the ETO family protein, Mtgr1. (A) Two-step immunoaffinity purification of Prdm14-associated proteins. FLAG–HA immunoprecipitations were performed from wild-type (wt) or FH-Prdm14 mESC extracts, followed by visualization of polypeptides by SDS-PAGE-silver stain and mass spectrometry identification. Polypeptides corresponding to Mtgr1 and Prdm14 are highlighted. (B) Reciprocal Mtgr1 and Prdm14 co-immunoprecipitations from FH-Prdm14 mESCs. (C, D) Identification of the Prdm14-Mtgr1 interaction regions. Co-immunoprecipitations were performed in HEK293 cells transfected with full-length V5-tagged Mtgr1 and distinct FH-Prdm14 constructs (C) or full-length FH-Prdm14 and distinct V5-Mtgr1 constructs (D), as indicated in the top diagrams; co-immunoprecipitated proteins were visualized by immunoblotting with α-HA (Prdm14) and α-V5 (Mtgr1) antibodies (top left and right panels, respectively). Tagged Prdm14 and Mtgr1 levels in the input extracts are shown in the bottom panels. (E) Recombinant biotinylated Prdm14 (pre-SET+SET) was immobilized on streptavidin-coated beads and incubated with recombinant Mtgr1 (NHR1), and changes in fluorescence (SAV-Dylight650) were measured. (F, G) Recombinant biotinylated Mtgr1 (NHR1) was immobilized on streptavidin-coated beads and incubated with (F) Prdm14 (pre-SET+SET) or (G) Prdm14 (SET) and changes in fluorescence (SAV-Dylight650) were measured. The error bars and the errors for the $K_D$ values are the standard deviation (n = 3). The curves show the best fit of the 1:1 binding model using the GraphPad software. * indicates non-specific bands. HEK, human embryonic kidney; Mtgr1, myeloid translocation gene related 1; MW, molecular weight marker; NHR1, nervy homology region 1. N.D., not determined.

The following source data and figure supplement are available for figure 1:

**Source data 1.** List of proteins recovered from the Prdm14 IP-MS experiment.

*Figure 1 continued on next page*

*Figure 1 continued*

**Figure supplement 1.** Expression levels of mRNAs encoding ETO proteins.

The obtained binding measurements yielded a dissociation constant ($K_D$) in the low nanomolar range (*Figure 1E and F*), which is consistent with a robust, direct interaction between the two proteins. On the other hand, the binding of the Prdm14 SET domain alone (residues 232–373) to the Mtgr1 NHR1 domain was barely detectable (*Figure 1G*), further supporting that both pre-SET and SET domains are required for the high affinity interaction with Mtgr1.

Altogether, our approach identified an ETO protein Mtgr1 as a novel, direct partner of Prdm14 in mESCs. While the ETO proteins, especially Mtg8 (a.k.a. ETO), have been studied in the context of acute myeloid leukemias (AML) (reviewed in *Hatlen et al., 2012*), their function in ESCs and during early embryogenesis has not been explored. Notably, all three ETO family members have the capacity to interact with Prdm14 (not shown), but the high expression of Mtgr1 in mESCs compared with Mtg8 and Mtg16 (*Figure 1—figure supplement 1*) likely accounts for the preferential recovery of Mtgr1 in our experiments and suggests that this family member may be most relevant in the context of mESCs. We therefore proceeded to explore the functional significance of the Prdm14–Mtgr1 interaction in mESC biology.

## Prdm14 and Mtgr1 co-occupy genomic targets

Prdm14 is a sequence-dependent DNA-binding protein that binds many genomic loci in mESCs, corresponding primarily to distal regulatory elements, whereas ETO proteins do not contain domains implicated in direct DNA sequence recognition (*Rossetti et al., 2004, 2008*). To examine whether Mtgr1 is brought to genomic targets occupied by Prdm14, we performed Mtgr1 ChIP coupled with high-throughput DNA sequencing (ChIP-seq) from wt mESCs, *FH-Prdm14* overexpressing mESCs, and as a control for antibody specificity, $Mtgr1^{-/-}$ mESCs (generation of which is described in more detail later), cultured for 5 days under serum+leukemia inhibitory factor (LIF) conditions. In parallel, we profiled Prdm14 occupancy by performing ChIP-seq analysis from *FH-Prdm14* cells, using an anti-HA antibody due to the unavailability of ChIP-grade Prdm14 antibodies. Overall, we identified ~ 8000 Mtgr1 peaks present in both *FH-Prdm14* and wt mESCs, but absent in $Mtgr1^{-/-}$ mESCs. These bound sites include loci known to be occupied and repressed by Prdm14 (e.g. near *Prdm14, Dnmt3b, Wnt8a, Peg10,* and targets of the FGFR pathway *Fgfr2* and *Shc1*; *Figure 2A*).

Generally, the genomic occupancies of Mtgr1 and Prdm14 were well correlated (correlation coefficient ~0.9, *Figure 2—figure supplement 1A*), and we have not been able to detect a substantial class of Prdm14-bound sites devoid of Mtgr1 occupancy (*Figure 2B*). Not surprisingly, the Prdm14 and Mtgr1 sites shared common functional ontologies, with enrichment for processes involved in embryonic development and cell differentiation (*Figure 2—figure supplement 1B*). Furthermore, the most highly enriched DNA sequence motif at Mtgr1-bound sites corresponded to the previously identified Prdm14 motif (*Figure 2C*). Interestingly, we noted that, at many targets, Mtgr1 binding was enhanced by Prdm14 overexpression (see tracks in *Figure 2A*, compare wt and *FH-Prdm14* ESC). This observation prompted us to quantitatively compare Mtgr1 ChIP-seq enrichments in wt ESCs and *FH-Prdm14* cells that are characterized by ~5-fold overexpression of Prdm14. We observed that Mtgr1 enrichments were higher in *FH-Prdm14* than in wt ESCs at most target sites, consistent with Prdm14-mediated recruitment of Mtgr1 to chromatin (*Figure 2D*). However, we also noticed that a subset of Mtgr1 sites was bound more weakly in *FH-Prdm14* cells than in wt ESCs (*Figure 2D*, examples shown in *Figure 2—figure supplement 2A*). The major distinction between these two populations was the presence of the Prdm14 sequence motif and Prdm14 occupancy at the sites where Mtgr1 binding was enhanced by Prdm14 overexpression, and lack of the Prdm14 sequence motif with low/no Prdm14 occupancy at the sites where Mtgr1 binding was diminished by Prdm14 overexpression (*Figure 2D*). Of note, at the Prdm14 motif-lacking sites, the most enriched sequence motifs corresponded to helix-loop-helix transcription factor recognition sites, suggesting that a TF from this family may be involved in mediating Mtgr1 binding at these sites (*Figure 2—figure supplement 2C*). Regardless, our results indicate that Prdm14 is sufficient to augment interaction of Mtgr1 with chromatin at its cognate binding sites and, at high levels, redirect it away from

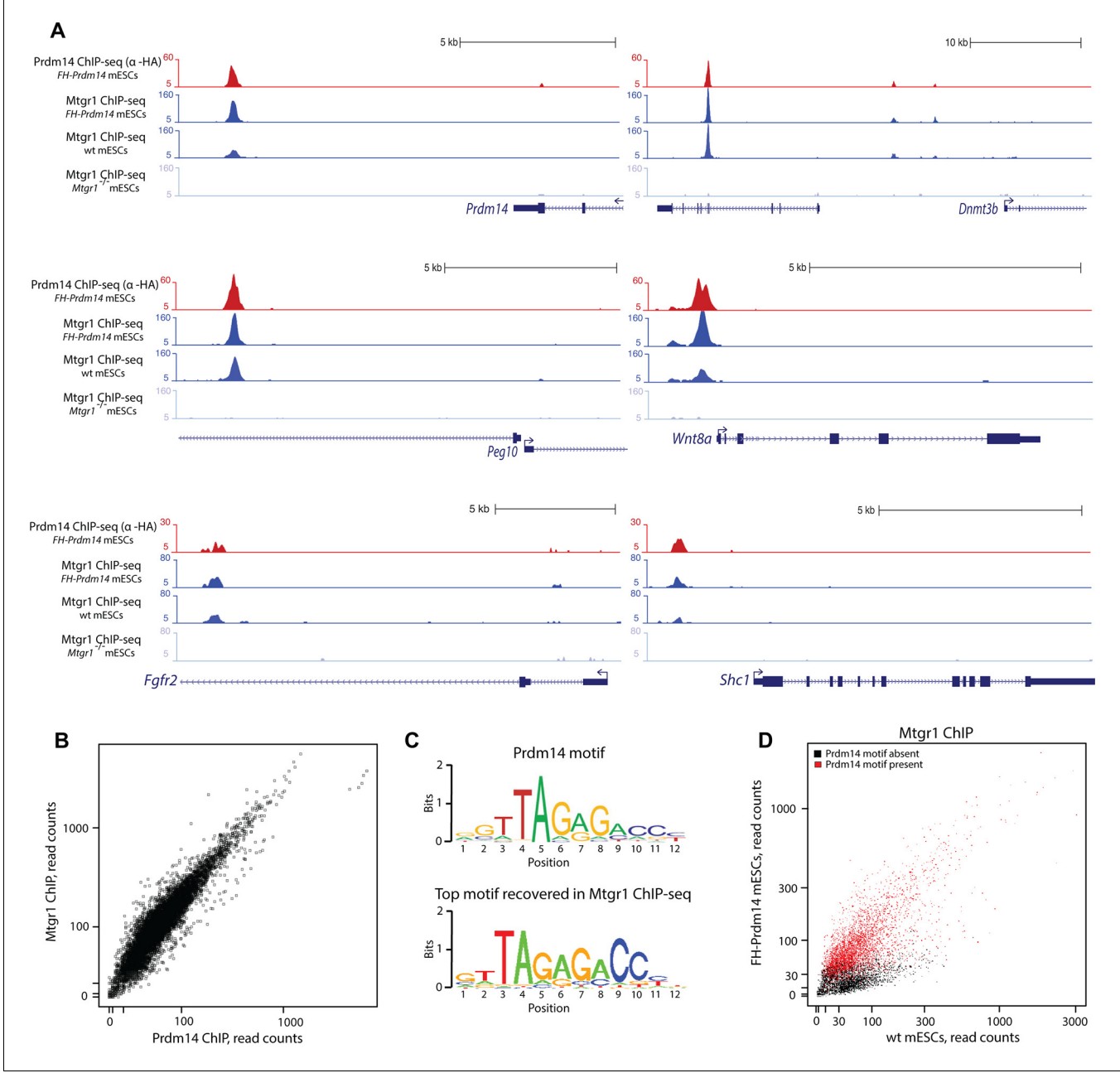

**Figure 2.** Prdm14 and Mtgr1 co-occupy genomic targets. (**A**) Prdm14 and Mtgr1 ChIP-seq enrichments at selected gene loci. Tracks represent sequence tag enrichments as determined by Quest software. (**B**) Scatter plot of Prdm14 and Mtgr1 genomic occupancies in FH-Prdm14 mESC line. (**C**) The top sequence motif recovered in Mtgr1 ChIP-seq corresponds to the Prdm14 motif, as defined previously (**Ma et al. 2011**). Logos for the consensus motifs were generated using SeqPos. (**D**) Scatter plot of Mtgr1 genomic occupancy in wt and FH-Prdm14 mESC lines. The plot is colored based on the presence of Prdm14 motif (red, motif is present, p-value $<10^{-3}$; black, motif is absent). ChIP-seq, chromatin immunoprecipitation with sequencing; mESC, mouse embryonic stem cell; Mtgr1, myeloid translocation gene related 1.

The following figure supplements are available for figure 2:

**Figure supplement 1.** Genomic occupancy of Mtgr1.

**Figure supplement 2.** Genomic occupancy of Mtgr1 at Prdm14 motif-lacking sites.

the motif-lacking sites. Thus, Prdm14 might be a limiting factor for Mtgr1 recruitment to chromatin. To test this notion further, we performed Mtgr1 ChIP-seq analysis from $Prdm14^{-/-}$ ESCs and generated average signal profiles at Prdm14 motif-containing and Prdm14 motif-lacking sites across all our Mtgr1 ChIP-seq datasets. We observed that at Prdm14 motif-containing sites, Mtgr1 binding is increased in FH-Prdm14 overexpressing cells and diminished (but not completely abrogated) in $Prdm14^{-/-}$ cells (*Figure 2—figure supplement 2B*, left panel). On the other hand, at Prdm14 motif-lacking sites, Mtgr1 binding is depleted by FH-Prdm14 overexpression, but it is also moderately affected in $Prdm14^{-/-}$ cells despite low/no Prdm14 binding at these sites, suggesting an indirect effect (*Figure 2—figure supplement 2B*, right panel). Altogether, these results are consistent with the Mtgr1 genomic occupancy being sensitive to the Prdm14 dosage (either loss or gain) at the Prdm14-motif containing sites. However, these results also demonstrate that even in the absence of Prdm14, some Mtgr1 binding remains at the motif-containing sites, suggesting partial redundancies in the recruitment mechanisms.

## Loss of Mtgr1 phenocopies requirement for Prdm14 in safeguarding pluripotency

Prdm14 has well-characterized roles in pluripotency and PGC formation, and if Mtgr1 is a key mediator of Prdm14's functions then the loss of Mtgr1 should impact these processes in a similar manner. To test this hypothesis, we used CRISPR-Cas9 with a guide RNA targeting the third exon of the *Mtgr1* gene to generate $Mtgr1^{-/-}$ mESCs, and verified the presence of the homozygous deletions and loss of the Mtgr1 protein in the three clonal lines selected for further analysis (*Figure 3—figure supplement 1*). As a reference for comparison, we also isolated and characterized two $Prdm14^{-/-}$ mESC lines by targeting the second exon of the *Prdm14* gene (*Figure 3—figure supplement 2*). Moreover, we reconstituted each of the $Mtgr1^{-/-}$ and $Prdm14^{-/-}$ cell lines with *FH-Mtgr1* or *FH-Prdm14* complementary DNA (cDNA), respectively, to generate 'rescue' cell lines and ensure specificity of the observed phenotypes. All aforementioned cell lines were isolated and maintained under the serum-free 2i+LIF conditions in which the major differentiation cues are inhibited and that support self-renewal even in the absence of Prdm14 (*Grabole et al., 2013*; *Yamaji et al., 2013*). After being transferred into standard serum+LIF growth conditions, the $Mtgr1^{-/-}$ lines exhibited changes in morphological appearance with less compact colonies, diminished cell–cell interactions and cell flattening, as previously reported for loss of Prdm14 in mESCs and reproduced here with our $Prdm14^{-/-}$ lines (*Ma et al., 2011*; *Yamaji et al., 2013*) (*Figure 3—figure supplement 3*). These features were not observed in wt mESCs or after rescue with the respective protein constructs (*Figure 3—figure supplement 3*).

Loss of Prdm14 has been shown to sensitize mESC to differentiation stimuli, resulting in upregulation of genes associated with epiblast and extraembryonic endoderm fates (*Ma et al., 2011*; *Yamaji et al., 2013*). To examine whether these molecular phenotypes are also observed upon loss of Mtgr1, we conducted RNA sequencing (RNA-seq) transcriptome analyses from wt mESCs, $Prdm14^{-/-}$ and $Mtgr1^{-/-}$ cell lines, and their respective rescue lines after transfer from 2i+LIF to serum+LIF conditions. As seen in $Prdm14^{-/-}$ mESCs, $Mtgr1^{-/-}$ mESCs showed upregulation of epiblast (e.g. *Fgf5, Dnmt3b, Oct6, Wnt8a*) and extraembryonic endoderm (e.g. *Krt19, Sparc, H19, Fgfr2*) markers, and downregulation of naïve pluripotency genes (e.g. *Esrrb, Zfp42, Tbx3, Tet2*), compared with either wt or *FH-Mtgr1* rescue mESCs (*Figure 3A*, *Figure 3—figure supplement 4A*).

Next, we identified genes showing the most variable expression across our datasets and visualized their expression changes in each of our RNA-seq datasets as a heatmap (*Figure 3B*). Most of the differentially expressed genes were concordantly upregulated in $Prdm14^{-/-}$ and $Mtgr1^{-/-}$ cells, compared with wt mESCs, in agreement with the proposed function of the Prdm14–Mtgr1 complex in gene repression. A more systematic comparison of all genes upregulated at least twofold upon loss of either Prdm14 or Mtgr1 revealed that while indeed, the majority of genes that are upregulated in either knockout are upregulated in both (*Figure 3—figure supplement 4B*, purple dots), a subset of transcripts is preferentially affected only in one of the knockouts (red or blue dots).

Importantly, in the *FH-Prdm14* or *FH-Mtgr1* reconstituted knockout cells the derepression defects were rescued (*Figure 3B*). Interestingly, while *FH-Mtgr1* cells showed expression patterns highly similar to that of wt mESCs cultured in serum, *FH-Prdm14* cells were more similar to the mESCs grown under 2i +LIF, despite being cultured in serum at the time of analysis (*Figure 3B*). Indeed, many of the expression differences observed between wt mESC grown in 2i versus serum were recapitulated

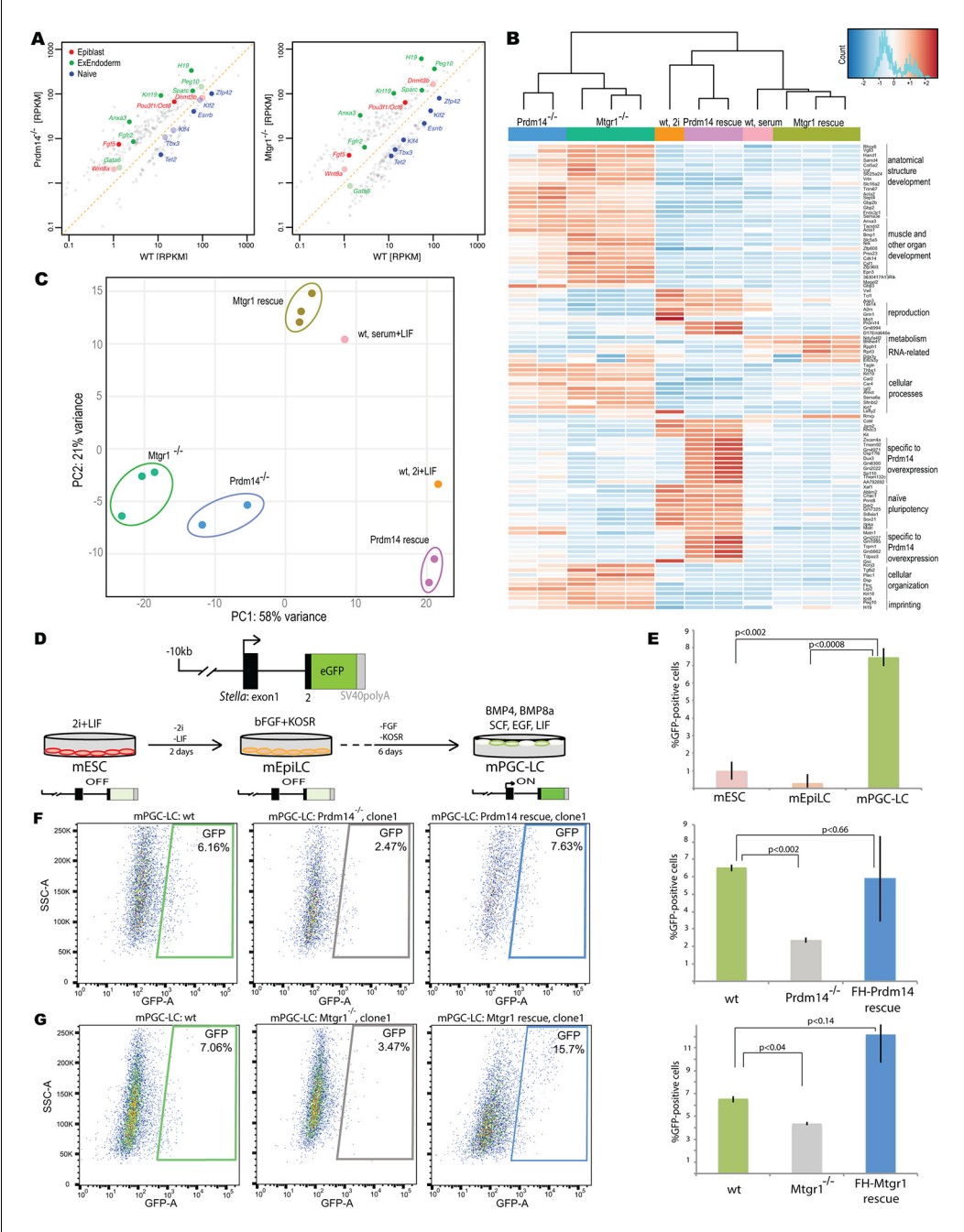

**Figure 3.** Loss of Mtgr1 phenocopies requirement for Prdm14 in safeguarding pluripotency and PGC induction. (**A**) RNA-seq from *Prdm14*[−/−] cells or *Mtgr1*[−/−] cells (y-axis) were compared to wt cells (x-axis) and expression values (RPKM) of all significantly changed transcripts were plotted. Select transcripts corresponding to those enriched in the post-implantation epiblast, extraembryonic endoderm or naïve pluripotent mESC are highlighted in red, green or blue, respectively; shaded colors indicate no significant difference. (**B**) Heatmap displaying top 100 variable genes between wt mESCs grown under naïve 2i+LIF or serum+LIF conditions, *Prdm14*[−/−] (2 clones) or *Mtgr1*[−/−] (3 clones) cells, and their respective rescue lines. Clustering represents sample divergence. (**C**) Principal component analysis on the same populations as in B. (**D**) mESC to mEpiLC transition followed by PGC-LC induction using defined media in cells containing *Stella*:GFP reporter. Schematic of the *Stella*:GFP transgene reporter that contains a 10kb 5′ upstream sequence and includes exon 1 and part of exon 2 fused in-frame with eGFP, followed by the SV40 polyadenylation sequence (*Payer et al. 2006*). The reporter is active in mPGC-LCs when *Stella* expression is activated. (**E**) Quantification of the GFP signal in wt cells during the mESC to mEpiLC and further to mPGC-LC transition. (**F**) FACS plots and gated quantification of GFP signal as a measure of mPGC-LC induction from wt cells, *Prdm14*[−/−] cells or Prdm14 rescue clones. (**G**) FACS plots and quantification of GFP signal as a measure of PGC-LC induction from wt cells, *Mtgr1*[−/−] cells or Mtgr1 rescue clones. FACS, fluorescence-activated cell sorting; GFP, green fluorescent protein; LIF, leukemia inhibitory factorm; EpiLCs, mouse epiblast-like

*Figure 3 continued on next page*

*Figure 3 continued*

cells; mESCs, mouse embryonic stem cell; mPGC-LCs, mouse primordial germ cell-like cells; Mtgr1, myeloid translocation gene related 1; RPKM, reads per kilobase of exon per million reads mapped; wt, wild-type.

The following figure supplements are available for figure 3:

**Figure supplement 1.** Generation of Mtgr1-null line in *Stella*:GFP mESCs using CRISPR-Cas9 system.

**Figure supplement 2.** Generation of Prdm14-null line in *Stella*:GFP mESCs using CRISPR-Cas9 system.

**Figure supplement 3.** Morphological changes associated with loss of Prdm14 or Mtgr1.

**Figure supplement 4.** Additional analyses of RNA-seq datasets.

**Figure supplement 5.** Expression of differentiation markers in embryoid bodies derived from wt or Mtgr1$^{-/-}$ ESCs.

**Figure supplement 6.** Loss of Mtgr1 results in defect in PGC-LC induction.

in *FH-Prdm14* mESCs (*Figure 3B*). Of note, a subset of transcripts was upregulated only in *FH-Prdm14* cells; many of those genes represent markers of the so-called 2-cell (2C) state (*Amano et al., 2013*; *Dan et al., 2013*; *Macfarlan et al., 2012*).

In our analysis of variably expressed genes, the *Prdm14*$^{-/-}$ and *Mtgr1*$^{-/-}$ cell lines clustered together, but separately from the respective rescue lines and wt mESCs (*Figure 3B*). These observations were further confirmed by the global comparisons of transcriptomes with principal component analysis (PCA), in which the *Prdm14*$^{-/-}$ and *Mtgr1*$^{-/-}$ cells were closest to each other and clustered away from the remaining cell lines (*Figure 3C*). Additionally, the PCA analysis corroborated higher similarity of *FH-Mtgr1* cells to wt mESCs grown in serum+LIF, and *FH-Prdm14* cells to wt mESCs grown under 2i+LIF conditions (*Figure 3C*). Given that: (i) the *FH-Prdm14* cell lines in our study express Prdm14 at levels ~5–6-fold higher than wt ESCs, (ii) Prdm14 has an autonomous DNA-binding activity, and (iii) Prdm14 overexpression can augment Mtgr1 recruitment to the target genes (as shown in *Figure 2*), we propose that expression changes observed in *FH-Prdm14* cell lines are associated with more robust repression of differentiation programs by Prdm14-Mtgr1 complex compared with wt cells and consequently, with the stabilization of the naïve pluripotency program. Similar gain-of-function effects are not observed in *FH-Mtgr1* cells, likely because Mtgr1 lacks the autonomous ability to access its genomic targets and, at least in mESCs, Prdm14 is limiting for its chromatin association. Altogether, our data uncover the function of Mtgr1 in safeguarding mESC pluripotency and demonstrate that loss of Mtgr1 phenocopies gene expression defects associated with Prdm14 deletion.

## Mtgr1 is required for PGC specification in vitro

Prdm14 is critical for the specification of PGCs from the post-implantation epiblast cells (*Magnúsdóttir and Surani, 2014*; *Yamaji et al., 2008*). To address whether Mtgr1 also plays a role in PGC formation, we used a previously established in vitro model in which naïve mESCs are first differentiated to a primed, post-implantation epiblast-like state (mEpiLCs) from which mouse primordial germ cell-like cells (mPGC-LCs) are then induced via addition of various cytokines (*Hayashi and Saitou, 2013*; *Hayashi et al., 2011*). The mPGC-LCs formation is monitored with the fluorescent reporter, *Stella*:GFP and quantified by fluorescence-activated cell sorting (FACS) analysis (*Figure 3D*). In vitro derived mPGC-LCs have been shown to be competent to differentiate to sperm/oocytes upon transplantation and produce viable, fertile offspring and this simple in vitro differentiation system is therefore considered a useful tool to study mechanisms underlying PGC specification (*Hayashi et al., 2011*; *Nakaki et al., 2013*). To examine the role of Prdm14 and Mtgr1 in the context of this model, we derived *Prdm14*$^{-/-}$ and *Mtgr1*$^{-/-}$ mESCs in the *Stella*:GFP reporter background; this reporter recapitulates endogenous *Stella* induction that occurs during the PGC formation (*Payer et al., 2006*).

In agreement with previous reports, differentiation of *Stella*:GFP mESCs consistently produced mPGC-LCs with an efficiency of ~7–8% on day 6 of differentiation, while low levels (~1%) of GFP-positive cells were detected in mESCs and further diminished upon mEpiLC formation (*Figure 3E*). In contrast, both *Prdm14*$^{-/-}$ and *Mtgr1*$^{-/-}$ mESCs showed significantly decreased efficiency of mPGC-LCs formation (*Figure 3F and G*, *Figure 3—figure supplement 5*). These defects were rescued by the re-introduction of *FH-Prdm14* or *FH-Mtgr1*, respectively (*Figure 3F and G*, *Figure 3—figure supplement 5*). These data suggest that Mtgr1, like Prdm14, is important for mouse PGC establishment. Significantly, loss of Mtgr1 does not result in general block in differentiation, as germ layer markers are expressed at comparable levels in embryoid bodies induced from *Mtgr1*$^{-/-}$ versus wt ESCs (*Figure 3—figure supplement 5*).

## Generation of renewable monobody reagents to study and inhibit Prdm14-Mtgr1 interaction

To develop new tools for understanding the function of Prdm14 and to aid structure determination, we generated designer binding proteins termed 'monobodies' recognizing the SET domain of human and mouse Prdm14, as a part of a larger project aimed at developing new reagents for controlling epigenetic regulatory proteins. Monobodies are small binding proteins (~10 kDa) generated from combinatorial phage-display libraries built on the antibody-like scaffold of the tenth human fibronectin type III domain (FN3; *Figure 4A*) (*Koide et al., 1998*, *2012b*). Monobodies can recognize their targets with high affinity and specificity and have a strong tendency to recognize functional binding sites in their target molecules including clefts and planar surfaces, and thus they often are potent inhibitors (*Koide et al., 2012a*; *Sha et al., 2013*; *Wojcik et al., 2010*). Furthermore, unlike antibodies whose folding depends on disulfide bond formation, monobodies are cysteine-free and thus functional when expressed under reducing environments such as the nucleus and cytoplasm. These attributes make monobodies particularly attractive as genetically encoded intracellular inhibitors.

We isolated Prdm14-binding monobodies from two combinatorial phage display libraries termed the 'loop' library and the 'side' library (*Koide et al., 2012a*). Following phage display selection, we performed gene shuffling for affinity maturation and further selection in the yeast display format to isolate clones with high affinity. We identified a total of 12 clones that bound to human PRDM14 (hPRDM14) with $K_D$ <100 nM as measured by yeast surface display (*Figure 4A*; *Figure 4—figure supplement 1*). Among them, we identified two clones, Mb(hPRDM14_S4) and Mb(hPRDM14_S14), that bound hPRDM14 (residues 238–487) at least five-fold stronger than the closest homologues, hPRDM6 (residues 194–405) and hPRDM12 (residues 60–229) (*Figure 4C*). We will use shorthand names, Mb(S4) and Mb(S14), for referring to them hereafter for brevity. Additionally, these two monobodies showed comparable binding to the mouse homologue of hPRDM14, Prdm14. Of note, hPRDM14 was able to substitute for the mouse Prdm14 in rescue of the Prdm14$^{-/-}$ ESC defects, suggesting both biochemical and biological conservation of function between mouse and humans (*Figure 4—figure supplement 2*).

We produced these two monobodies as purified proteins for further characterization. Consistent with analysis using yeast surface display, these purified monobodies showed high affinity with $K_D$ <50 nM to both hPRDM14 and Prdm14 in bead-based assays (*Figure 4B*). We then examined whether these monobodies inhibited the interaction of Prdm14 with Mtgr1. Mb(S14) potently competed against the binding of Mtgr1 to Prdm14 but Mb(S4) did not (*Figure 4D*). This result suggests that the two monobodies bind to distinct surfaces of Prdm14 and the epitope for Mb(S14) overlaps with and therefore occludes the Mtgr1-binding surface (*Figure 4D*).

We next tested whetherif these monobodies can immunoprecipitate Prdm14 from mESC lysates. Mb(S4) captured Prdm14 and co-immunoprecipitated vast majority of Mtgr1 from the *FH-Prdm14* cell extracts (*Figure 4E*). On the other hand, Mb(S14) captured lower levels of Prdm14 , in agreement with its competition for the same binding surface as Mtgr1 (*Figure 4E*). Since PRC2 complex has been previously reported to associate with Prdm14 (*Chan et al., 2013*; *Payer et al., 2013*; *Yamaji et al., 2013*), we also looked for the presence of Suz12, a PRC2 component. We did not detect immunoprecipitated Suz12 in the elution fraction for either of the two monobodies. Next, we used monobodies to precipitate endogenous Prdm14 from wt ESCs. Immunoblot analysis with α-Mtgr1 antibody showed that Mb(S4) monobody, which does not disrupt Prdm14-Mtgr1 interaction, recovered endogenous Mtgr1 (and was able to deplete most of it from the extract, *Figure 4—figure*

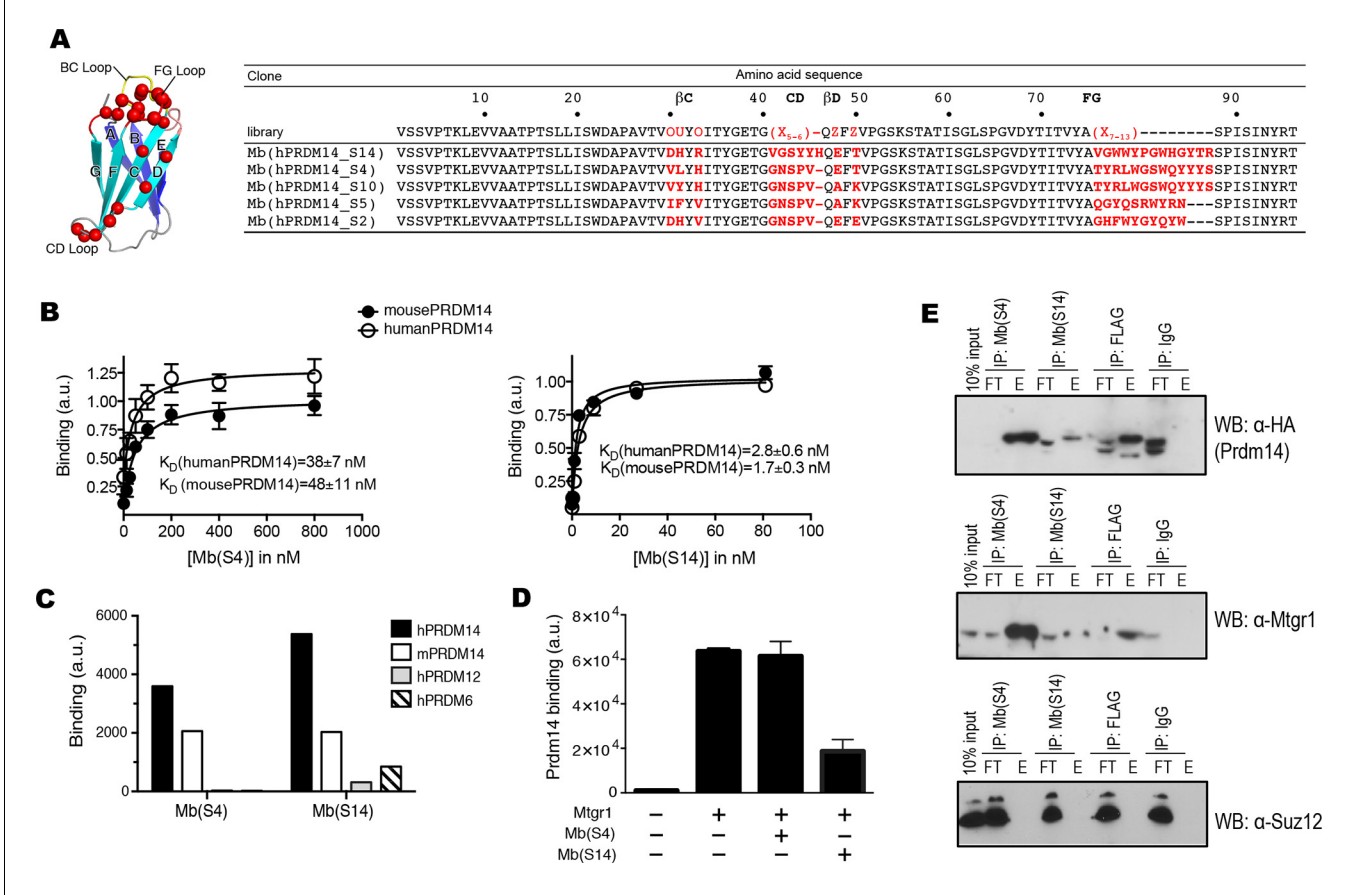

**Figure 4.** Generation of PRDM14-binding monobodies. (**A**) Schematic of the monobody scaffold. The β strands and loops are labeled and the diversified residues are marked as red spheres. The amino acid sequences of the monobody library and monobody clones. In the library designs, 'X' denotes a mixture of 30% Tyr, 15% Ser, 10% Gly, 5% Phe, 5% Trp, and 2.5% each of all the other amino acids except for Cys; 'O', a mixture of Asn, Asp, His, Ile, Leu, Phe, Tyr, and Val; 'U', a mixture of His, Leu, Phe, and Tyr; 'Z', a mixture of Ala, Glu, Lys, and Thr (*Koide et al. 2012*). A hyphen indicates a deletion. (**B**) Titration curves of Mb(hPRDM14_S4) and Mb(hPRDM14_S14) to human PRDM14 and mouse Prdm14. The error bars are the standard deviation (n = 3). The curves show the best fit of the 1:1 binding model. (**C**) Binding of Mb(S4) and Mb(S14) expressed on yeast surface to 50 nM of hPRDM14 and its homologues, mouse Prdm14, human PRDM12 and human PRDM6. (**D**) Competitive binding assay for Mtgr1 and monobodies. Binding of 10 nM Mtgr1 to biotinylated Prdm14 immobilized on streptavidin coated M280 beads in the absence and presence of 500 nM purified monobodies, Mb(S4) and Mb(S14). (**E**) Co-immunoprecipitation of FLAG–HA tagged Prdm14 expressed in mESC using Mb(S4), Mb(S14), α-FLAG M2 antibody or a negative control antibody ('IgG'). Antibodies used for Western blotting are indicated with the blots. E, elution; FT, flow through; Mtgr1, myeloid translocation gene related 1.

The following figure supplements are available for figure 4:

**Figure supplement 1.** Sequences of monobodies selected against human PRDM14 and their $K_D$ values to hPRDM14 measured in yeast display format.

**Figure supplement 2.** Human PRDM14 can substitute for the mouse Prdm14 in mESCs.

**Figure supplement 3.** Monobody affinity pulldown of the endogenous Prdm14 protein.

*supplement 3A and B*). In addition, we performed Prdm14 Mb(S4)-precipitation/mass spec analysis from wt ESCs, which readily detected Prdm14- and Mtgr1-originating peptides, but did not recover any other PRDM proteins confirming a high specificity of this monobody (*Figure 4—figure supplement 3B and C*). Common to monobodies generated to recognize native, folded proteins, neither Mb(S4) nor Mb(S14) detected denatured Prdm14 in immunoblotting (not shown). Overall, we report here the generation of the first recombinant affinity reagents targeting two distinct sites of Prdm14 SET domain.

## Monobody- and fusion-assisted crystal structure determination of the Prdm14-Mtgr1 complex

To understand how Prdm14 and Mtgr1 interact at the atomic level, we attempted crystallization of the Prdm14-Mtgr1 complex. However, aggregation of both proteins resulted in low yields of the complex suitable for crystallization. To overcome this problem, we designed a fusion construct of Prdm14 and Mtgr1 in which the two proteins were linked with a ten-residue linker (GSSGSSGS), a common strategy for stabilizing heterodimers (*Ernst et al., 2014*; *Kobe et al., 2015*; *Reddy Chichili et al., 2013*; *Zhou et al., 2015*). To confirm that the linker did not distort the Prdm14–Mtgr1 complex, we performed a series of experiments. The fusion protein had the same retention time on size-exclusion chromatography as the unlinked complex (*Figure 5—figure supplement 1A*), indicating that the linker did not alter the stoichiometry of the complex. We then compared the fusion and the unlinked complex using solution nuclear magnetic resonance (NMR) spectroscopy. Most of the cross peaks in the heteronuclear single quantum coherence (HSQC) spectrum of $^{15}$N-Prdm14 in complex with unlabeled Mtgr1 (where we observe signals only from $^{15}$N-Prdm14) overlapped with those in the HSQC spectrum of $^{15}$N-labeled Prdm14-linker-Mtgr1 (where we observe signals from the entire fusion protein including both Prdm14 and Mtgr1) (*Figure 5—figure supplement 1B*). The large number of overlapping peaks in the two spectra strongly suggests that the Prdm14 protein takes on nearly identical average conformation in the unlinked complex and the fusion protein (*Figure 5—figure supplement 1B*). Furthermore, the fusion protein and the unlinked complex had the same affinity to Mb(S4), indicating that the linker did not distort the Prdm14 epitope for the monobody (*Figure 5—figure supplement 1C*). This fusion construct allowed us to overcome the aggregation problem, but it still yielded no crystals in crystallization trials using over 500 conditions.

We then used Mb(S4), the monobody that did not inhibit the Prdm14-Mtgr1 interaction, as a crystallization chaperone. Monobodies, like antibody fragments, often facilitate the crystallization of otherwise recalcitrant systems (*Koide, 2009*; *Stockbridge et al., 2015*). The addition of Mb(S4) readily led to crystallization of Prdm14-linker-Mtgr1, and we determined its structure to a resolution of 3.06 Å through single wavelength anomalous diffraction (SAD) phasing using selenomethionine-labeled crystals (*Figure 5A*; *Table 1*; *Figure 5—figure supplement 2A*). The crystallized complex had two Prdm14-linker-Mtgr1/Mb(S4) complexes in the asymmetric unit. As expected, the monobody bound exclusively to Prdm14, burying 604 Å$^2$ surface areas, a similar interface size to other monobody/target complexes (*Figure 5—figure supplement 2B, E*) (*Gilbreth et al., 2008*; *Wojcik et al., 2010*). In the crystal, monobody–monobody interactions facilitated crystal contacts via face-to-face interactions of the β-sheet surfaces (not involved in Prdm14 interaction), illustrating the importance of Mb(S4) as a crystallization chaperone for this complex (*Figure 5—figure supplement 2D*).

The Prdm14 SET domain (residues 240–356) in the crystal structure is flanked by pre-SET and post-SET regions. The crystallization construct includes residues 184–239 that precede the SET domain (pre-SET) and residues 357–373 following the SET domain (post-SET). Unlike many SET domain-containing proteins, Prdm14 does not have cysteine-rich Zn finger domains adjacent to the SET domain, commonly termed pre-SET and post-SET domains. Thus, in the absence of well-defined domains, we refer to these adjacent segments as pre-SET and post-SET regions. The crystal structure of the mouse Prdm14 SET domain is very similar to that of the hPRDM12 SET domain, the closest human homologue of Prdm14 (PDB ID 3EP0; Cα RMSD=0.99; *Figure 5B*). The Prdm14 SET domain in our structure has a total of nine β-strands (β1-β9) arranged in three antiparallel β-sheets with a short 3$_{10}$ helix (η1) inserted between β6 and β7. The pre-SET region in our construct has a short helix at the N-terminus followed by a long structurally disordered region a part of which (residues 217–239) has no detectable electron density even at 0.5 σ contour levels (2Fo-Fc). The residues that constitute the post-SET region at the C-terminus to the SET domain are arranged in an antiparallel beta sheet (β10 and β11). Overall, the structural features of the PR/SET domain in Prdm14 show no major differences with other PR/SET domains in Prdm proteins.

The Mtgr1 NHR1 domain (also called the TAFH domain) is highly conserved in the ETO family. The Mtgr1 NHR1 domain in the crystal structure contains four well-defined α-helices arranged in a bundle (αA-αD; *Figure 5C*). Currently, three NMR structures for the NHR1 domain of human MTG8, another member of the ETO family, are available (*Park et al., 2009*; *Plevin et al., 2006*; *Wei et al., 2007*). Mtgr1 in the crystal structure has almost identical topology as the average solution NMR

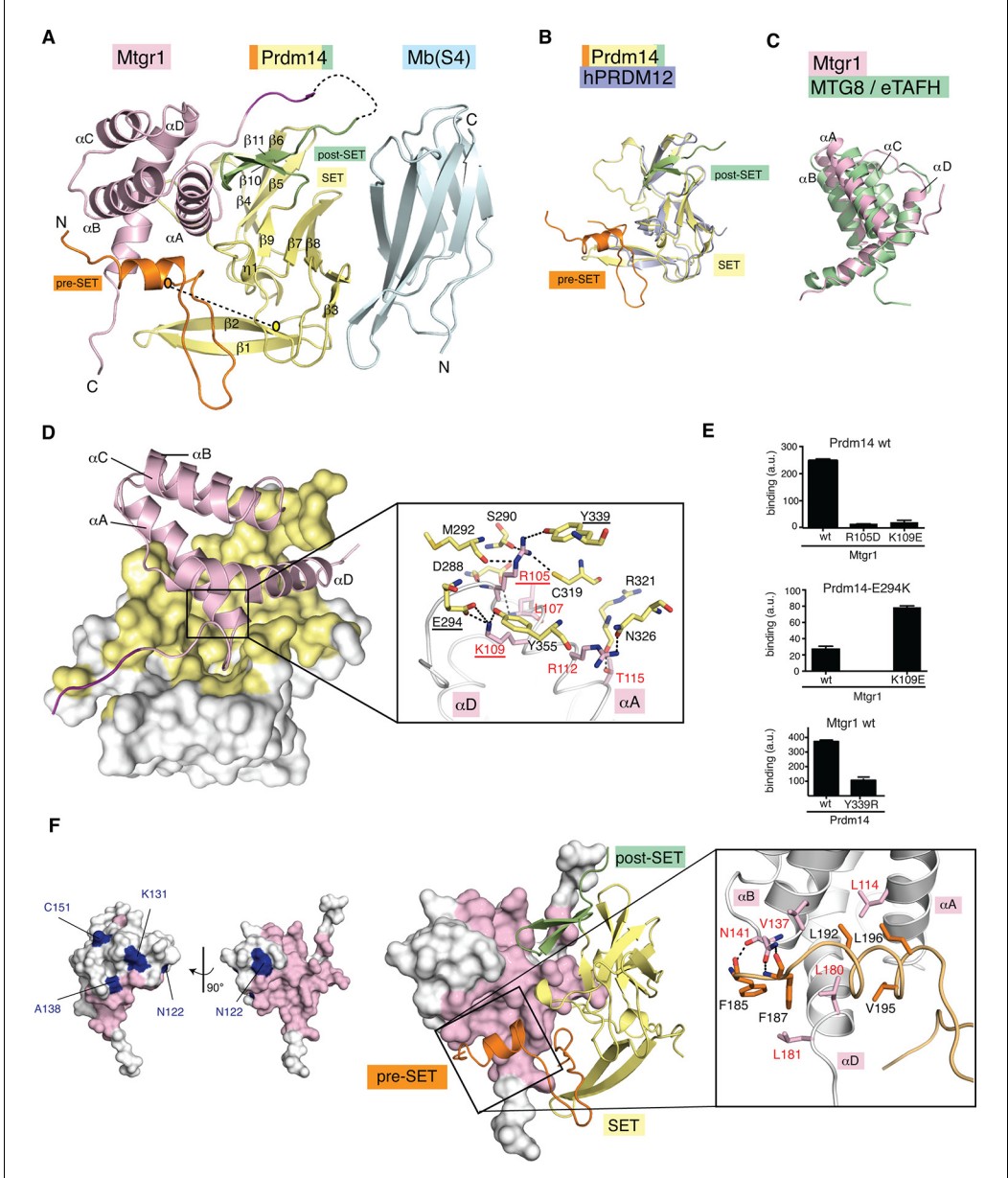

**Figure 5.** Crystal structure of the Prdm14–Mtgr1 complex. (**A**) The overall structure of the Prdm14-linker–Mtgr1/Mb(S4) complex. Missing residues are shown with dotted lines. The pre-SET region (in orange), SET domain (in yellow) and post-SET region (in green) of Prdm14 are shown. The Mtgr1 helices are marked for clarity (αA–αD). (**B**) Superposition of the Prdm14 crystal structure with the crystal structure of hPRDM12 (PDB ID 3EP0). (**C**) Superposition of the Mtgr1 crystal structure with the NMR structure of the MTG8 NHR1 (eTAFH) domain (PDB ID 2KNH). (**D**) Prdm14–Mtgr1 interface. Prdm14 is shown in white surface representation with the interacting residues in yellow (Top). Mtgr1 is shown in cartoon representation in pink color. In the detailed view, Prdm14 residues are marked in red and Mtgr1 residues are marked in black. Salt bridges and hydrogen bonds between Prdm14 and Mtgr1 are shown in dotted lines. Residues that were mutated based on the structure are underlined. (**E**) Binding of wt, E294K and Y339R Prdm14 (residues 184–373) to immobilized wt Mtgr1 (top) and Mtgr1(K109E) (bottom) in a bead-based assay. (**F**) Mtgr1 is shown in white surface representation with interacting residues in pink (left). Non-identical residues between Mtgr1 and the other ETO proteins are shown in blue (left). The detailed view shows the interaction of Prdm14 residues N-terminal to the SET domain (pre-SET) with Mtgr1 (right). Prdm14 residues are labeled in red and Mtgr1 residues in black. Mtgr1, myeloid translocation gene related 1; NMR, nuclear magnetic resonance.

The following figure supplements are available for figure 5:

**Figure supplement 1.** Inclusion of a linker does not affect the Prdm14–Mtgr1 interaction.

**Figure supplement 2.** Structural features of the Prdm14-linker–Mtgr1/Mb (S4) complex.

*Figure 5 continued on next page*

*Figure 5 continued*

**Figure supplement 3.** Comparison of the structure of the Prdm14–linker-Mtgr1 complex with that of the Prdm9- histone H3 peptide-S-adenosyl-L-homocysteine (AdoHcy) complex.

structure of the NHR1 domain from MTG8 (ETO) in complex with a stabilizing peptide (PDB ID 2KNH; Cα RMSD=1.77 Å; *Figure 5C*). Similar structural features and high sequence identity for the NHR1 domain in ETO proteins explain the pull-down of Mtg8 and Mtg16 in the FH-Prdm14 purifications (*Figure 5F* (left); *Figure 1A*; *Figure 1—source data 1*).

## Mtgr1 contacts with the SET domain and pre-SET region of Prdm14

The Mtgr1-Prdm14 interaction interface buries 2180 Å$^2$, a large interface but still within the observed range for high-affinity protein–protein interaction interfaces with a low nanomolar $K_D$ value (*Lo Conte et al., 1999*). We observe that Mtgr1 interacts with both SET domain and pre-SET region of Prdm14 (*Figure 5A*), as was predicted from our earlier interaction mapping co-immunoprecipitation experiments (*Figure 1C*). The interactions between the Prdm14 SET domain and Mtgr1, mediated primarily by αA and αD, contribute 65% of the total buried surface area. The substantial interaction interface between the pre-SET region and Mtgr1 (35% of the interface) rationalizes the importance of the pre-SET region in the Prdm14-Mtgr1 interaction (*Figure 1C*).

Several features in the interface are notable. Arg105 located at the N-terminal end of Mtgr1 αA sits in a pocket formed by Ser290, Met292, Cys319 and Tyr339 of Prdm14 and its guanidinium moiety makes hydrogen bonds with the side chains of all of these residues (*Figure 5D*, *Figure 5—figure supplement 2C*). Mutating Mtgr1 Arg105 to Asp resulted in a loss of detectable binding (*Figure 5E*), and replacing Prdm14 Tyr339 of the pocket with Arg substantially reduced binding (*Figure 5E*). In addition, Lys109 in Mtgr1 αA interacts with Glu294, Tyr355 and Pro366 of Prdm14 SET domain. Lys109 in Mtgr1 forms a salt bridge with Glu294 from β5 of the Prdm14 SET domain (*Figure 5D*). Mutating either of these residues led to a loss of detectable binding (*Figure 5E*). Remarkably, charge reversal of this ionic interaction, that is, the combination of Mtgr1 K109E and Prdm14 E294K restored binding, in agreement with the salt bridge formation across the binding interface in solution (*Figure 5E* middle). These mutation experiments together supports the authenticity of the binding interface observed in the crystal structure and identified critical electrostatic interactions.

The Mtgr1 interaction with Prdm14 pre-SET region involves residues from Mtgr1 helix αA, αB and αD (*Figure 5A*). Several Leu and Val residues from the helix in the pre-SET region contribute to hydrophobic interactions with residues in Mtgr1 helices (*Figure 5F*). In addition, Mtgr1 Asn141 forms hydrogen bonds with residues Phe185 and Phe187 from the helix in the pre-SET region. These results further reinforce the observation that Prdm14 utilizes both pre-SET region and SET domain to interact with Mtgr1.

## Association of Prdm14 with Mtgr1 is required for mESC maintenance and PGC-LC formation

Given strong association between Prdm14 and Mtgr1, as well as phenotypic similarities upon loss of either protein, we next examined whether the lack of Prdm14–Mtgr1 interaction would have a biological effect on mESC gene expression and the efficiency of PGC-LC induction. To this end, we first confirmed that the point mutations designed based on the crystal structure (*Figure 5D and E*) also disrupted the interaction between the full-length proteins in cells, and that the combined charge reversal mutations restored the interaction (*Figure 6A*). Then, we reconstituted *Stella*:GFP Prdm14$^{-/-}$ mESCs with cDNAs encoding either wt or mutant (E294K or Y339R) FH-Prdm14 and confirmed that all three proteins were expressed at similar levels (*Figure 6B*). Next, we used RNA-seq to compare gene expression patterns of these mESCs after transition from 2i+LIF to serum+LIF culture (*Figure 6C*). In the Prdm14$^{-/-}$ cells reconstituted with mutant Prdm14, we noted elevated expression of genes associated with differentiation to epiblast and extraembryonic endoderm, and diminished expression of naïve pluripotency genes (with an exception of *Tet2*), similar to attenuated expression patterns (albeit not to the same degree) as we observed upon loss of Prdm14 (*Figure 6C*, compare to *Figure 3A*).

**Table 1.** Data collection, phasing and refinement statistics for Prdm14-linker-Mtgr1/Mb(S4) complex crystals.

| | Native | SeMet SAD |
|---|---|---|
| **Data collection** | | |
| Beamline | APS 19ID | APS 19ID |
| Space group | $P4_32_12$ | $P4_32_12$ |
| *Cell dimensions* | | |
| *a, b, c* (Å) | 106.8,106.8,180.7 | 106.9, 106.9,180.9 |
| a, b, g (°) | 90,90,90 | 90,90,90 |
| | | *Peak* |
| Wavelength | 0.97918 Å | 0.97918 Å |
| Resolution (Å) | 37.7–3.05 (3.16–3.05) | 50–3.18 (3.23–3.18) |
| $R_{pim}$ | 0.024 (0.482) | 0.022 (0.315) |
| I / σI | 30.0 (1.4) | 52.4 (2.0) |
| Completeness (%) | 100 (100) | 100 (100) |
| Redundancy | 20.5 (19.5) | 86.2 (45.2) |
| **Refinement** | | |
| Resolution (Å) | 37.7–3.05 (3.16–3.05) | |
| No. of unique reflections | 20631 (2013) | |
| $R_{work}$ / $R_{free}$ | 0.189/0.250 | |
| No. atoms | 5476 | |
| Protein | 5476 | |
| Ligand/ion | 0 | |
| Water | 0 | |
| *B*-factors | 114.3 | |
| Protein | 114.3 | |
| Ligand/ion | 0 | |
| Water | 0 | |
| R.m.s deviations | | |
| Bond lengths (Å) | 0.006 | |
| Bond angles (°) | 1.13 | |
| *Ramachandran statistics* | | |
| Favorable | 95.8 | |
| Allowed | 4.1 | |
| Outliers | 0.1 | |

Principal component and clustering based on differential gene expression further support the notion that Prdm14 E294K lines show hypomorphic expression profile, which falls in between the Prdm14$^{-/-}$ ESCs and those reconstituted with wt Prdm14 (*Figure 6—figure supplement 1*). Finally, the efficiency of mPGC-LCs formation from mESC reconstituted with Prdm14 mutants (E294K, Y339R) was significantly diminished compared to cells rescued with the wt Prdm14 to almost the same degree as in *Prdm14*$^{-/-}$ cells (*Figure 6D*).

Our results thus demonstrate that association of Prdm14 with Mtgr1 is required for mediating its functions in pluripotency and germ cell formation. Given that one of the monobodies we developed, Mb(S14), binds to Prdm14 competitively with Mtgr1, we hypothesized that this reagent can be utilized to inhibit Prdm14 function in living cells or organisms in a highly controlled manner. To provide a proof-of-principle for such strategy, we engineered piggyBac doxycycline-inducible constructs

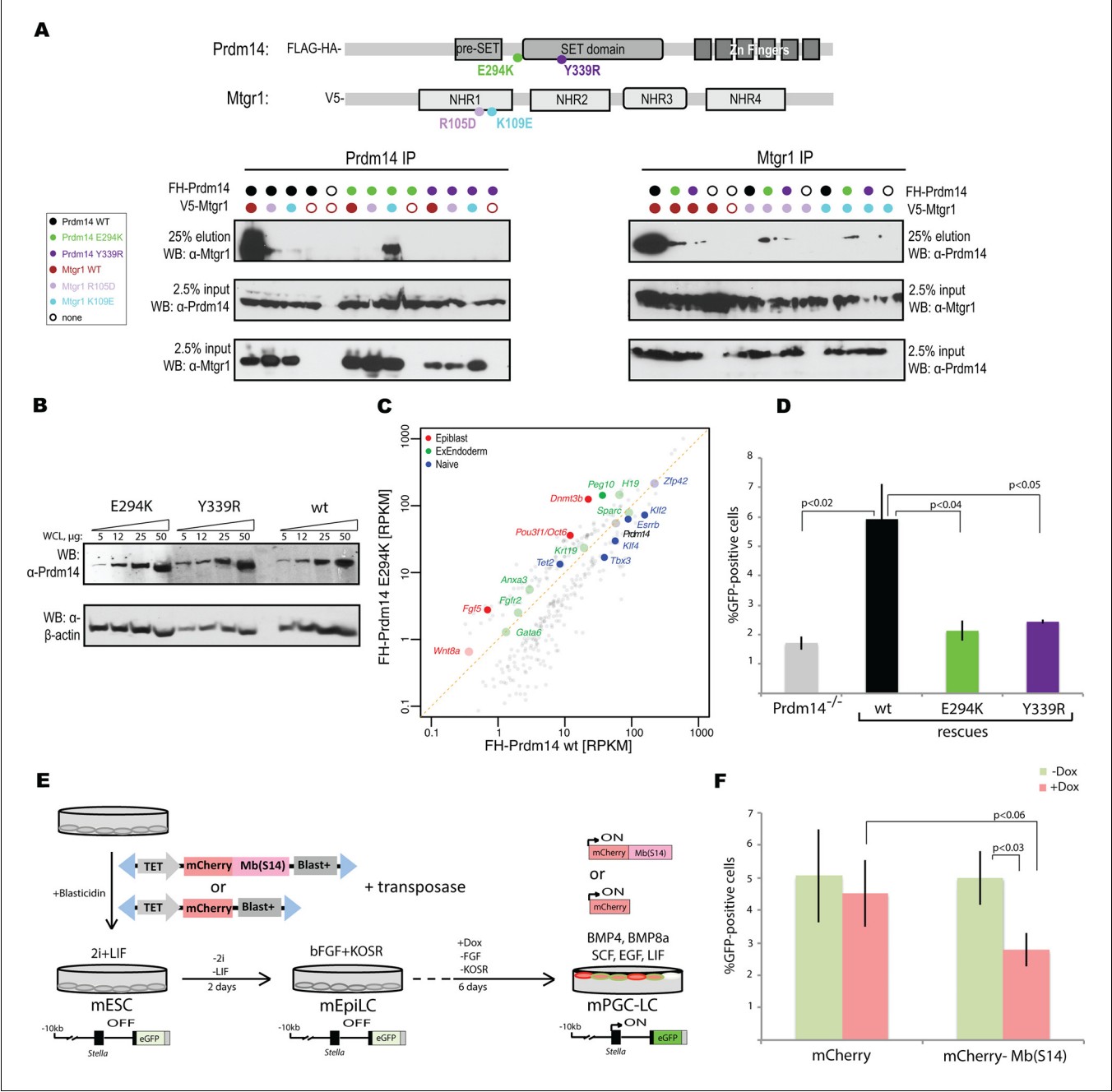

**Figure 6.** Inhibition of Prdm14–Mtgr1 interaction affects stem cell maintenance and PGC-LC induction. (**A**) Single amino-acid substitutions at the interaction surface abrogate Prdm14–Mtgr1 association in cells. Indicated full length V5-Mtgr1 wt or single point mutant proteins were introduced to HEK293 cells and co-immunoprecipitated with full length FH-Prdm14 wt or single mutant protein as indicated in the diagram. Please note rescue of the association when the combination of Mtgr1 K109E and Prdm14 E294K mutants is tested. (**B**) Western blot showing protein expression levels of wt, E294K and Y339R FH-Prdm14 protein in lysates from *Prdm14*$^{-/-}$ mESCs cells reconstituted with the respective transgenes. (**C**) RNA-seq from *Prdm14*$^{-/-}$ cells reconstituted with wt Prdm14 protein (x axis) were compared to *Prdm14*$^{-/-}$ cells reconstituted with E294K Prdm14 mutant (y axis) and expression values (RPKM) of all significantly changed transcripts were plotted. The transcripts of specific genes are highlighted in red, green, blue or black as indicated; shaded colors indicate no significant difference. (**D**) Quantification of GFP signal as a measure of mPGC-LC induction from *Prdm14*$^{-/-}$ cells and *Prdm14*$^{-/-}$ cells reconstituted with transgenes encoding wt, E294K, or Y339R Prdm14 protein. (**E**) Schematics of the piggyBac transposon-based reporter system used to create dual reporter lines. mESC line was transfected with either dox-inducible mCherry construct or dox-inducible mCherry-Mb(S14) fusion protein. The lines were selected using blasticidin and three populations were tested further for their competency to form mPGC-LCs. mESC to mEpiLC transition followed by mPGC-LC transition using defined media in cells containing *Stella*:GFP reporter (lower panel). Doxycycline was added after mEpiLC stage. (**F**) Quantification of GFP signal as a measure of mPGC-LC induction from mCherry population of cells and mCherry-Mb

*Figure 6 continued on next page*

Figure 6 continued

(S14) population of cells with and without addition of doxycycline. mEpiLCs, mouse epiblast-like cells; mESCs, mouse embryonic stem cell; mPGC-LCs, mouse primordial germ cell-like cells; GFP, green fluorescent protein; Mtgr1, myeloid translocation gene related 1; RNA-seq, RNA sequencing; RPKM, reads per kilobase of exon per million reads mapped; WCL, whole cell lysates; wt, wild type.
The following figure supplements are available for figure 6:

**Figure supplement 1.** Gene expression analyses of Prdm14$^{-/-}$ cells reconstituted with wt or E294K Prdm14.

**Figure supplement 2.** Inhibition of Prdm14–Mtgr1 interaction affects germ cell development.

encoding mCherry-Mb(S14) fusion or, as a control, mCherry alone, and introduced them into *Stella*: GFP reporter mESCs (*Figure 6E*). Next, we induced mPGC-LC formation from mESCs in the absence or presence of doxycycline (added during the induction of mPGC-LCs from mEpiLCs) to activate monobody expression. We observed consistent reduction in PGC-LC formation efficiency in cells expressing the mCherry-Mb(S14) fusion protein, as compared with the same cell population without addition of doxycycline or to cells expressing mCherry alone (*Figure 6F*, *Figure 6—figure supplement 1*). A more moderate effect of the monobody (compared with the Prdm14 E294K mutation, *Figure 6D*) is probably due to the fact that this monobody needs to compete against the high-affinity interaction between Prdm14 and Mtgr1. In addition, we noticed a short half-life of the mCherry-Mb(S14) fusion protein (data not shown), which may contribute to the moderate effect but could also facilitate inhibition with high temporal resolution in the future. Thus, Mb(S14) represents a novel tool that can be utilized to perturb Prdm14 function in living cells during a dynamic biological process such as the PGC induction and can be further modified in the future for addressing particular questions using standard protein-engineering technologies.

## Discussion

Our study identified the ETO family co-repressor Mtgr1 as a new regulator of mESC identity, which facilitates molecular functions of Prdm14 through direct binding to its pre-SET/SET region. Therefore, although no evidence has been previously found for the catalytic activity of the Prdm14 SET domain, our data demonstrate that this domain is nonetheless essential for Prdm14 function by mediating the interaction with its key partner, Mtgr1. Prdm14 and Mtgr1 co-occupy distal regulatory regions of many target genes linked to differentiation, DNA methylation and chromatin modification, consistent with their tight interaction. Loss of either protein results in upregulation of a subset of these target genes and a gradual loss of mESC self-renewal. However, similar to what has been observed previously for Prdm14, Mtgr1 is not required for maintenance of mESC under 2i+LIF. This can be explained by the fact that the Fgf/Erk pathway, a major signaling cascade driving both epiblast and extraembryonic endoderm differentiation (to which loss of Prdm14/Mtgr1 sensitizes cells), is inhibited under these conditions (*Nichols and Smith, 2009*).

Concordant upregulation of gene transcripts in *Prdm14*$^{-/-}$ or *Mtgr1*$^{-/-}$ cells compared with wt mESCs supports function of the Prdm14–Mtgr1 complex in gene silencing and agrees with the reported role of Mtgr1 as an HDAC-recruiting co-repressor (*Rossetti et al., 2004*). Notably, we detected HDACs 1–3 in our Prdm14 immunoprecipitates, and HDAC3 and other NCoR1 repressive complex components in our purifications with the Mb(S4) monobody, suggesting that the Prdm14–Mtgr1-dependent repression may indeed be facilitated by histone deacetylation and that Mtgr1 is required for recruiting HDACs to Prdm14-binding loci. Furthermore, although a subset of genomic sites occupied by Mtgr1 occurs at Prdm14 motif-lacking sites, the high similarity of transcriptional changes observed upon loss of either protein suggests that the major impact of Mtgr1 on gene expression and cell identity of mESCs is in the context of its association with Prdm14. Moreover, our data suggest that Prdm14 can not only guide, but through change in its levels, quantitatively tune the degree of interaction of ubiquitously expressed Mtgr1 with chromatin.

Our results from the in vitro PGC-LC formation model strongly suggest that in addition to its role in mESCs, Mtgr1 is also a critical mediator of the Prdm14 function in germline development. Three lines of evidence support this notion: (i) Mtgr1 deletion; (ii) single point mutations in Prdm14

disrupting association with Mtgr1; and (iii) expression of a monobody targeting the Prdm14–Mtgr1 interaction surface, all of which hinder PGC-LC induction in vitro. It is therefore surprising that a *Mtgr1* mouse knockout strain has been reported as viable and fertile (*Amann et al., 2005*). However, it remains unclear whether this strain represents a true loss-of-function, because in the mouse targeting strategy, the first six exons are preserved (encoding amino acids 1–316, which span the Prdm14 interaction region and NHR2 domain involved in dimerization and association with mSin3/HDAC complex) (*Rossetti et al., 2004*). In light of our findings and aforementioned caveats, we suggest that the in vivo function of Mtgr1 in the germline should be revisited. It is also possible, however, that requirement for Mtgr1 in germ cell development in vivo can be fully or partially compensated by the other ETO proteins. Because residues that form the Prdm14 binding interface in Mtgr1 are conserved among the ETO family members, we expect that the other ETO proteins would bind directly to the Prdm14 pre-SET/SET regions and could therefore compensate for the absence of Mtgr1 function.

Our study provides novel insights into how pre-SET and SET regions might mediate high affinity protein–protein interactions. While many structures of the catalytic SET domains have been obtained with their substrates (typically, histone tails), to the best of our knowledge, our study represents the first structural analysis of the SET domain acting as a module for a high affinity protein–protein interaction. Interestingly, a comparison of our structure with that of the Prdm9–AdoHcy–histone peptide complex indicates that the surface of the Prdm14 pre-SET and SET regions engaged in interaction with Mtgr1 overlaps with surfaces other Prdm proteins use for binding to their histone peptide substrate (*Figure 5—figure supplement 3B*; [*Wu et al., 2013*]). Indeed, Mtgr1 Lys109 is in close proximity to Tyr355 that corresponds to the catalytic Tyr based on the consensus SET domain sequence (*Smith and Denu, 2009*). In Prdm9, the catalytic tyrosine Tyr356 along with Tyr276 and Tyr341 form the Lys4me2 binding pocket and are critical for catalytic activity (*Wu et al., 2013*). In Prdm14, the side chain of Tyr355 flips over (with respect to the conformation of Tyr357 in Prdm9) and interacts with K109 of Mtgr1. The location of K109 is distinct from that of Lys4me2 in Prdm9 (*Figure 5—figure supplement 3C*). In addition, the high affinity of the Prdm14–Mtgr1 interaction strongly suggests that Mtgr1 would be a poor substrate, given that a substrate needs to be released after catalysis for efficient enzyme reaction. We also note that His211 and Ala212 in the pre-SET region of Prdm14 occupy the S-adenosyl-L-methionine (SAM)-binding site (*Figure 5—figure supplement 3A*) and thus, at least in the context of the presented structure, preclude the binding of this cofactor necessary for methylation. Taken together, our results suggest that Mtgr1 Lys109 is unlikely to be an actual substrate for Prdm14-mediated methylation. Consistent with this notion, radioactive in vitro methyltransferase assays with recombinant Prdm14 and Mtgr1 proteins, their respective interaction mutants, as well as histone substrates, all failed to yield methyltransferase activity (not shown). However, we cannot exclude the possibility that under presently unknown conditions, such activity can ultimately be found. Regardless, our comparisons suggest that substrate-binding surfaces can, in some SET domain proteins, be co-opted for mediating high affinity protein–protein interactions, which may provide a molecular explanation as to why such surfaces are typically highly conserved even in SET domain proteins with no apparent catalytic activity.

Lastly, aberrant reactivation of the *PRDM14* locus is associated with a variety of human cancers, and mice overexpressing Prdm14 in blood cells develop early-onset T-cell acute lymphoblastic leukemia (T-ALL) (*Carofino et al., 2013*). We speculate that the oncogenic function of Prdm14 may be mediated by the formation of a complex with Mtgr1, which is broadly expressed and thus readily available for association in many different organ systems, or perhaps with other ETO proteins using the same interface. If this proves to be the case, inhibition of the Mtgr1-interaction surface on Prdm14 could be an attractive target for therapy as Prdm14 expression is restricted under non-pathological conditions to the preimplantation embryo and PGCs, reducing the risk of off-target effects on normal somatic tissues. Thus, the monobodies directed to Prdm14 generated in this study will be powerful tools for testing the 'druggability' of the Prdm14–Mtgr1 interaction. Indeed, we have introduced monobodies into CML cells and demonstrated the potential druggability of a domain interface in Bcr-Abl (*Grebien et al., 2011*). In addition, the crystal structure shows that Mb(S4) binds to a distinct surface of Prdm14. Because monobodies usually bind to functional sites on target proteins, we hypothesize that the Mb(S4) epitope may also be important for Prdm14 function, which will be a subject of future research. We emphasize that the genetically encoded monobodies are portable

tools, as they can be readily introduced into different cells via transfection or viral transduction. This attribute should facilitate the investigation of Prdm14 functions in diverse contexts.

## Materials and methods

### Cell culture, line derivation and embryoid body formation

Stable lines expressing tagged Prdm14 were established from single colonies by transducing LF2 mESCs with FH-Prdm14 pTrip lentivirus for 24 hr, followed by selection with neomycin as described previously (*Ma et al., 2011*); these lines were used for immunoprecipitations followed by mass spectrometry experiments. *Stella:*GFP cells were a gift from A. Surani and were used for all other experiments and further genetic manipulations, unless indicated otherwise. This line contains a transgene spanning 10 kb upstream of the *Stella* transcriptional start site, exon1, intron1, and part of exon2, followed by *eGFP* fused in-frame and SV40 polyadenylation sequence. *Stella:*GFP line acts as a transcriptional reporter and has been previously shown to faithfully recapitulate endogenous *Stella* expression in mouse and mark PGCs as early as E7.5 (*Payer et al., 2006*). Stable double reporter lines (*Stella:*GFP and mCherry) were created by transfecting piggyBac Tet-On expression plasmid controlled by rtTA and doxycycline (mCherry alone or mCherry fused to Mb(S14)) with transposase and selected with blasticidin for 7 days. Single clones were picked and expanded further.

For maintenance, all mouse ESC lines were grown in so-called '2i+LIF' medium that is serum-free N2B27-based medium supplemented with MEK inhibitor PD0325901 (0.8 μM) and GSK3β inhibitor CHIR99021 (3.3 μM) in tissue culture (TC) dishes pretreated with 7.5 μg/ml polyl-ornithine (Sigma) and 5 μg/ml laminine (BD) (*Hayashi and Saitou, 2013*; *Hayashi et al., 2011*). For ChIP-seq, RNA-seq and immunoprecipitation experiments, *Stella:*GFP mESC lines and FH-Prdm14 derivatives were cultured for 5 days in feeder-free conditions in Dulbecco's Modified Eagle Medium (DMEM)-high glucose medium (DMEM/high glucose; HyClone) containing 15% (v/v) fetal bovine serum, 1 mM glutamine, penicillin/streptomycin, 0.1mM nonessential amino acids, 0.1 mM 2-mercaptoethanol and supplemented with LIF (serum+LIF conditions).

To form embryoid bodies, Stella:GFP mESCs line of interest was maintained in a 15 cm Petri dish with serum culture medium without LIF, to allow aggregation as a hanging drop with 400 cells per drop (360 drops per mESC line). Cells were cultured for 4 days and half of them were collected for further analysis by RT-qPCR, while the other half was transferred into non-adherent Petri dish and allowed to grow for another 4 days (8 days total) before RT-qPCR analysis.

### EpiLC and PGC-LC differentiation

To induce mEpiLC differentiation, mESC were washed with phosphate-buffered saline, trypsinized, and strained. A total of about 100,000 cells per one well of 12-well plate were plated on TC dishes pretreated with 5 μg/ml fibronectin (Millipore) in N2B27-based medium supplemented with 1% KSR (Invitrogen) and 12 μg/ml bFGF (Peprotech). The mPGC-LCs were induced similarly to what has been described previously for 6 days (*Hayashi and Saitou, 2013*; *Hayashi et al., 2011*). Specifically, 1000–2000 mEpiLC cells were aggregated in a hanging drop in a serum-free medium (GMEM, Invitrogen) supplemented with 15% KSR, 0.1 mM NEAA, 1 mM sodium pyruvate, 0.1 mM 2-mercaptoethanol, penicillin/streptomycin, 1 mM glutamine and cytokines 500 ng/ml BMP4 (R&D Systems), 500 ng/ml BMP8a (R&D Systems), 100 ng/ml SCF (R&D systems), 50 ng/ml EGF(R&D systems), LIF. Where indicated doxycycline was added at the point of mPGC-LCs induction from mEpiLCs.

### RT-qPCR expression analysis

Total RNA was isolated with Trizol and afterwards treated with turbo DNase. For reverse transcription of mRNAs, we used 1 μg of DNAse digested RNA, random hexamer primers (5×TransAmp Buffer, Bioline) and reverse transcriptase (Bioline) in 20-μl reaction volume. qPCR analyses were carried out with SensiFAST SYBR No-ROX kit (Bioline) on LightCycler 480 II qPCR machine (Roche).

### RNA-seq

RNAs from at least two independent biological replicates of indicated cell lines were extracted with Trizol (Invitrogen), following the manufacturer's recommendations. Ten micrograms of total RNA were subjected to two rounds purification using Dynaloligo-dT beads (Invitrogen). Purified RNA was

fragmented with 10× fragmentation buffer (Ambion) and used for first-strand cDNA synthesis, using random hexamer primers (Invitrogen) and SuperScript II enzyme (Invitrogen). Second strand cDNA was obtained by adding RNaseH (Invitrogen) and DNA Pol I (New England BioLabs). The resulting double-stranded cDNA was used for Illumina library preparation and sequenced with Illumina Genome Analyzer. Following library preparation, samples were pooled and sequenced on an Illumina NextSeq instrument using 76 base-pair single-end reads on a NextSeq high output kit (Illumina) or HiSeq instrument using 51 base-pair single-end reads.

## Chromatin immunoprecipitation with sequencing

ChIP assays were performed from $10^7$ mESC per experiment, according to previously described protocol with slight modification (*Rada-Iglesias et al., 2011*). Briefly, cells were crosslinked with 1% formaldehyde for 10 min at room temperature and the reaction was quenched by glycine at a final concentration of 0.125 M. Chromatin was sonicated to an average size of 0.5–2 kb, using Bioruptor (Diagenode). A total of 5 µg of antibody was added to the sonicated chromatin and incubated overnight at 4°C. Subsequently, 50 µl of protein G Dynal magnetic beads were added to the ChIP reactions and incubated for ~4 hr at 4°C. Magnetic beads were washed and chromatin eluted, followed by reversal of crosslinks and DNA purification. ChIP DNA was dissolved in water. ChIP-seq and input libraries were prepared according to Illumina protocol and sequenced using Illumina Genome Analyzer. Following library preparation, samples were pooled and sequenced on an Illumina NextSeq instrument using 76 base-pair single-end reads on a NextSeq high output kit (Illumina).

## RNA-seq analysis

Quality of FASTQ files was assessed using FastQC software. Raw sequencing reads were aligned using Tophat against mm9 genomic index and with refseq gene models as available at illumina.com ftp site. Aligned reads were converted to counts for every gene using HTSeq and gene counts were further analyzed using R and DESeq2 package (*Love et al., 2014*). For the scatter plot we identified significantly affected transcripts and plotted RPKMs of transcripts significantly (q<0.05) affected by Prdm14 or Mtgr1 loss. We compiled from published literature a set of official gene symbols for representative marker genes characteristic for epiblast, extraembryonic endoderm and naïve lineages.

The heatmap was created by looking at top 100 genes with the highest variance across samples (topVarGenes). We looked at the amount by which each gene deviates in a specific sample from the gene's average across all samples; thus, we centered and scaled each gene's values across samples and then plotted a heatmap. To visualize sample-to-sample distances between different lines we used PCA, plotPCA function within DESeq2 package in R on the rlog-transformed counts.

## ChIP-seq analysis

Quality of FASTQ files was assessed using FastQC software. ChIPseq peak calls were done with MACS2 callpeak with default settings (https://pypi.python.org/pypi/MACS2). Superset of intervals was created by merging summits from all calls using mean shift algorithm with 300 bp bandwidth. The modal peaks were extended ± 300 bp and read coverage was calculated with bedtools. Regions with outlier counts in negative controls were excluded from further analysis.

## DNA sequence motif analysis

Motifs enriched in Mtgr1 ChIP peaks were obtained with SeqPos (*He et al., 2010*), using a set of 1884 coordinates for top Mtgr1 peaks with signal higher in Prdm14 overexpressing cells than in wt cells (Prdm14-dependent) and 1721 Mtgr1 peaks with signal higher in wt cell than in the Prdm14 overexpression background (Prdm14-independent). ChIP regions containing the MTGR1/Prdm14 motif were identified with FIMO with $P$ value cutoff set to 0.001. In wt mESCs 64% of Prdm14-dependent sites contain the Prdm14 motif and 16% of Prdm14-independent sites contain the Prdm14 motif (p-value cutoff at 0.0003). This corresponds to odds ratio 9.84 and $p<<10^{-16}$ in Fisher's exact test.

## CRISPR/Cas9 targeting

Wt Cas9 plasmid pX330 was obtained from Addgene. The sgRNAs sequences were designed using Zhang Lab website (http://crispr.mit.edu/). Guide for Prdm14 was in exon 2

(CGCCGCCGAGGACCAAATTTTGG, score 95) and guide for Mtgr1 was in exon 3 (GACTCTCGTTC-TAGCCTTGGTGG, score 78). Note that guides against exons 1, 2, and 3 within Mtgr1 were designed as well as nickase version of Cas9 was used, but only aforementioned guide within exon3 produced mutations that resulted in the loss of protein. *Stella:*GFP mESC line was transfected with the desired sgRNA in pX330 plasmid together with piggyBac mCherry (transient transfection) using Lipofectamine 2000 (Life Technologies) according to the manufacturer's instruction manual. Forty-eight hours post-transfection, we did single-cell sorting into 96-well plates on mCherry-positive cells, assuming that these cells got transfected with both mCherry and pX330 plasmids. The colonies that arose from single cells were screened for the presence of the deletion. The target sequence was amplified by PCR with specific primers from genomic DNA. We then picked a restriction enzyme close to the PAM sequence that upon mutation of the sequence would not be able to cut. For Prdm14, we used *Pfl*MI restriction enzyme, and for Mtgr1, we used *Sty*I restriction enzyme. Clones that could not be digested were further analyzed by doing PCR with specific primers from cDNA and subsequently Sanger sequencing. The results were analyzed with Sequencher 5.1 software and TIDE (*Brinkman et al., 2014*). Clones that were confirmed to have a mutation in cDNA were further validated for the presence of protein using Western blotting.

## Fluorescence-activated cell sorting

Cells carrying *Stella:*GFP reporter were used to monitor the efficiency of mPGC-LCs formation after 6 days of differentiation. Cells carrying dual reporter constructs (mCherry-S14 Mb and *Stella:*GFP) were PGC-LCs induced for 6 days with doxycycline after which the cells were analyzed. Differentiation was carried out in hanging drops as described. The cells were trypsinized, strained through a 30µm cell strainer and analyzed on an LSR Fortessa Analyzer (BD), data were analyzed further using FlowJo. For statistical analysis, Student's t test was used to compare two normally distributed data sets. The analysis was done in R using unpaired t-test and paired t-test for the same cell population before and after doxycycline treatment. $p < 0.05$ was considered to be statistically significant.

## Immunoprecipitation

Dignam nuclear extracts from mESCs were prepared as previously described (*Peng et al., 2009*). For immunoprecipitations, monobodies or antibodies that were used are listed in the antibody section below. Typically, 50–100 pmol of monobody and 50 µl of pre-washed M280 dynabeads were used per immunoprecipitation. For immunoprecipitations performed using antibodies, we used 5 µg of antibody and 75 µl of ProteinG-sepharose (Sigma) beads per immunoprecipitation. In double-step IP we first used FLAG M2-beads with peptide elution followed by incubation with HA antibody. If the immunoprecipitation was followed by mass spectrometry peptide identification, then the eluant was run on the one-dimensional SDS-PAGE gel, fixed and excised.

## Western blotting

Cells were lysed with radioimmunoprecipitation assay buffer (50 mM Tris HCl pH 8, 300 mM NaCl, 1% Triton X-100, 0.5% sodium deoxycholate, 0.1% SDS, 1mM ethylenediaminetetraacetic acid [EDTA]) containing protease inhibitors (Roche tablet) and 1 mM DTT. The protein concentration was estimated with Bradford reagent (Bio-Rad) and equal or indicated amounts of protein were run on 8% SDS-PAGE gels and transferred to nitrocellulose membranes. Antibodies used in this study are listed in the antibody section.

## Protein identification by mass spectrometry

In gel digestion was performed as previously reported (*Shevchenko, et al., 2007*) with the addition of Protease Max for increased peptide and protein solubility. The extracted peptides were dried using a speed vac and reconstituted in mobile phase A. The ultra performance liquid chromatography (UPLC) was a Waters M-class where mobile phase A was 0.2% formic acid, 5% dimethyl sulfoxide (DMSO), 94.8% water and mobile phase B was 0.2% formic acid, 5% DMSO, 94.8% acetonitrile. The UPLC was run at 300 nl/min from 4% mobile phase B to 35% mobile phase B followed by a wash and re-equilibration step. The mass spectrometer was an Orbitrap Fusion mass spectrometer set to acquire in a data dependent fashion to optimize cycle time and fragment ion acquisition. The RAW data was searched using Byonic against the Uniprot mouse database downloaded on 09/29/2015.

The fixed modifications were Cys. propionamide, an the variable Met. oxidation, Asp. deamidation and N-terminal modifications. The data was filtered and presented at a 1% false discovery rate.

## Protein expression and purification

An expression vector for human PRDM14 (residues 238–487) with an N-terminal biotin-acceptor tag and C-terminal His$_6$ tag based on the p28BIOH-LIC vector (GenBank accession EF442785) was kindly provided by Susanne Gräslund and Cheryl Arrowsmith (Structural Genomics Consortium). The genes for Prdm14 (residues 184–373) and Mtgr1 (residues 98–206) were assembled using synthetic oligonucleotides and cloned in the pHBT vector that adds an N-terminal His$_6$ tag followed biotin-acceptor tag and a TEV cleavage site (*Sha et al., 2013*). The Mtgr1 construct in this vector contained a H200K mutation located outside the NHR1/TAFH domain due to a cloning artifact. For binding assays, all proteins were expressed in BL21 (DE3) cells containing pBirACm plasmid (Avidity) in the presence of 50 μM biotin to produce biotinylated proteins. The Prdm14-Mtgr1 fusion protein was designed to have a GSSGSSGS linker separating Prdm14 (residues 184–373) and Mtgr1 (98–206). The DNA sequences for these genes have been deposited to the GenBank.

All proteins were expressed as His$_6$-tagged proteins as described. Proteins were purified using Ni-Sepharose gravity flow columns (GE Healthcare) and the monodispersity of these proteins was assessed by size-exclusion chromatography. For crystallization, the fusion tags were removed using tobacco etch virus (TEV) protease cleavage, and the tags were removed using Ni-Sepharose columns.

## Phage display and yeast display-based selection

The method of selecting target specific monobodies from phage and yeast display libraries has been previously described (*Koide et al., 2012a*, *2012b*). Two monobody libraries ('loop' and 'side') were used to generate monobodies with diverse binding modes (*Koide et al., 2012a*). Each of these libraries contains approximately 10 billion unique monobody clones in which 16–26 residues are diversified using highly tailored amino acid combinations (*Gilbreth and Koide, 2012*; *Koide et al., 2012a*). Four rounds of phage display selection were performed using target concentrations of 100 nM, 100 nM, 75 nM and 50 nM. Streptavidin-coated magnetic beads (Streptavidin MagneSphere Paramagnetic Particles; Promega, Z5481/2) were used for immobilizing the target and captured phages were eluted with 0.1M Gly-HCl, pH 2.1. After gene shuffling among the selected clones within the enriched population (*Koide et al., 2012a*), the monobody-coding genes were transferred into a yeast display vector. We performed library selection by yeast surface display using magnetic beads in the first round followed by two rounds of FACS-based selection. Binding assay for testing the affinity and specificity of individual monobody clones was performed using yeast surface display as described previously (*Sha et al., 2013*).

## Bead-based binding assays

The general methods for bead-based assays have been described (*Nishikori et al., 2012*). In the assay, streptavidin-coated Dynabeads M280 beads (Invitrogen) at 20 μg/ml were incubated with 5 nM biotinylated target protein diluted in BSS/EDTA/DTT buffer (50 mM Tris–HCl, 150 mM NaCl, pH 8, 1 mg/ml bovine serum albumin, 1 mM EDTA, 0.1 mM DTT) for 30 min. The remaining free biotin-binding sites of streptavidin on the M280 beads were blocked with 5 μM free biotin for 30 min. Ten microliters of the target-immobilized beads were transferred to the wells of a 96-well filter plate (MultiScreen HTS HV, 0.45 μm, Millipore), drained using a vacuum manifold (MultiScreen HTS Vacuum Manifold, Millipore), and washed with 100 μl of BSS/EDTA/DTT buffer. Next, a biotinylated protein (biotinylation of the proteins was checked by their ability to bind to streptavidin beads) to be tested at various concentrations was added to individual wells and incubated for 30 min with gentle shaking. Then the wells of the filter plate were washed twice with 150 μl of the buffer, 20 μl of 10 μg/ml SAV-Dylight650 (ThermoFisher) in the buffer was added to the wells, and the plate incubated with shaking for 30 min. The wells were washed again and the beads resuspended in 140 μl buffer and analyzed using a Guava EasyCyte 6/l flow cytometer (Millipore).

## Crystallization of the Prdm14-linker–Mtgr1/Mb(S4) complex

Purified Prdm14-linker-Mtgr1 and Mb(S4) were mixed in the molar ratio of 1.0:1.3 and the complex was purified using a Superdex 75 16/600 size exclusion chromatography column (GE Healthcare) in 25 mM Tris pH 8.0, 100 mM NaCl, 0.2 mM TCEP. The protein complex was then concentrated to a final concentration of 15 mg/ml. Initial crystallization screening of ~ 500 conditions was carried out in 96-well plates using the hanging-drop vapor-diffusion method with a crystallization robot (Mosquito, TTP Labtech). Crystals used for data collection were obtained in 17% PEG3350 and 8% Tascimate pH 5.5, and were cryoprotected in 1:1 mix of Paratone and Paraffin oil and flash-cooled in liquid nitrogen prior to data collection. The Prdm14-linker–Mtgr1 protein was labeled with SeMet as described (*Doublié, 1997*), purified as a complex with the monobody and crystallized in a similar manner as the native proteins.

## Data collection, structure determination and refinement

X-ray diffraction data were collected at Beamline 19ID of the Advanced Photon Source (Argonne National Laboratory, Chicago, IL, USA) (*Table 1*). The data were indexed and integrated using HKL3000 (*Minor et al., 2006*). Molecular replacement using PHASER (*McCoy et al., 2007*) and MOLREP (*Vagin and Teplyakov, 1997*) with an hPrdm12 structure (PDB ID 3EP0) and the mono-body backbone (PDB 3UYO) did not have sufficient scattering power to *generate* a solution with a signal-to-noise ratio that is high *enough* to be identified. Thus, we determined the structure through single-wavelength Se anomalous dispersion experiment. A total of eight Se sites were identified and refined using Autosol (*Terwilliger et al., 2009*), resulting in an overall figure of merit of 0.45 and Z-score of 43.1. These phases were then used against the SAD data for model building in phenix.auto-build (*Adams et al., 2010*). Iterative model building and refinement were done using the programs COOT (*Emsley and Cowtan, 2004*) and PHENIX (*Adams et al., 2010*).The structure refined from SAD data was later refined against the higher resolution native data at a 3.05 Å resolution. The final structures were analyzed using Procheck and Molprobity (*Davis et al., 2004*). Figures were made using Pymol (DeLano, 2002). The structure has 100% residues in the allowed regions of the Rama-chandran plot with no outliers. The Molprobity score (2.16) is above average for structures refined at comparable resolutions.

## NMR spectroscopy

Uniformly $^{15}$N-labeled Prdm14-linker-Mtgr1 and $^{15}$N-labeled Prdm14 were prepared by growing bacterial cells in M9 minimal media supplemented with $^{15}$N-labeled ammonium sulphate (0.8 g l$^{-1}$, Cambridge Isotope Laboratories). The labeled proteins were purified in the same manner as the unlabeled proteins described above. $^{15}$N-labeled Prdm14 in complex with unlabeled $^{14}$N-Mtgr1 was purified by gel filtration chromatography. NMR data was collected at 30°C on a 600 MHz Bruker AVANCE III Spectrometer. The samples used for data collection contained 50–200 μM protein in 50 mM Tris-Cl buffer pH 8.0 containing 150 mM NaCl and 0.2 mM TCEP supplemented with 10% D$_2$O. All spectra were processed by the NMRPipe software (*Delaglio, et al., 1995*) and analyzed using SPARKY (*Goddard and Kneller*).

## Antibodies

Antibodies for Mtgr1 (Western and IP, ab53363, lot GR56108-2,4; ChIP, ab96161), V5tag (Western and IP, ab27671, lot GR186433-4), HA (ChIP, ab9110, lot GR146572-8) were from Abcam, and Suz12 (IP, 04–046) from Millipore. HA antibody (Western and IP, H3663), anti-Flag M2 agarose beads (A2220), M2 Flag antibody (Western, F1804) were from Sigma and dynabeads M280 streptavidin (11205D) were from Life Technologies.

## Acknowledgements

We would like to thank members of the Wysocka lab for insightful comments and critical editing of the manuscript. We thank A Surani for *Stella*:GFP mESCs, J Coller, D Wagh, and X Ji at Stanford Functional Genomics Facility for assistance with high-throughput sequencing, L Picton for help with NMR data collection, S Ginell and Y Kim at the Advanced Photon Source Sector 19ID for beamline access and assistance during data collection and processing, and S Gräslund and C Arrowsmith for

providing expression vectors. This work was supported by the Howard Hughes Medical Institute funds, National Institutes of Health grants R01GM112720 (JW), R01DA036887 and R01GM090324 (SK), P30 CA014599 (the University of Chicago Comprehensive Cancer Center, SK), and Canadian Institutes for Health Research post-doctoral fellowship (NN). This research used resources of the APS, a U.S. Department of Energy (DOE) Office of Science User Facility operated for the DOE Office of Science by Argonne National Laboratory under Contract No. DE-AC02-06CH11357.

## Additional information

### Competing interests

AK, SK: SK and AK are inventors on a patent application filed by the University of Chicago that covers monobody library design (US 13/813,409). The monobodies described in this work are available from SK under a material transfer agreement with the University of Chicago. The other authors declare that no competing interests exist.

### Funding

| Funder | Grant reference number | Author |
|---|---|---|
| National Institutes of Health | Research Project Grant R01 | Shohei Koide<br>Joanna Wysocka |
| Canadian Institutes of Health Research | Post-doctoral fellowship | Nataliya Nady |
| University of Chicago | Support Grant | Shohei Koide |
| Howard Hughes Medical Institute | Support | Joanna Wysocka |

The funders had no role in study design, data collection and interpretation, or the decision to submit the work for publication.

### Author contributions

NN, Designed and performed biochemical and cellular studies of Prdm14-Mtgr1 interaction presented in Figures 1, 2, 3, 4E and 6, with exceptions of Figure 1A, E-G, Conception and design, Acquisition of data, Analysis and interpretation of data, Drafting or revising the article, Contributed unpublished essential data or reagents; AG, Performed biophysical, structural and protein-engineering studies presented in Figures 1E-G, 4A-D and 5, Acquisition of data, Analysis and interpretation of data, Drafting or revising the article; ZM, Performed IP-MS experiments in Fig. 1A and done preliminary work on Mtgr1-Prdm14 functional association, Acquisition of data; TS, Contributed to analyses of RNA-seq and ChIP-seq datasets and provided experimental advice, Analysis and interpretation of data; AK, Provided input on experiments in Figure 4, Analysis and interpretation of data; SK, Guided experimental design and data interpretation; wrote the manuscript, Conception and design, Analysis and interpretation of data, Drafting or revising the article; JW, Guided experimental design and data interpretation; Wrote manuscript, Conception and design, Analysis and interpretation of data, Drafting or revising the article

## Additional files

### Major datasets

The following datasets were generated:

| Author(s) | Year | Dataset title | Dataset URL | Database, license, and accessibility information |
|---|---|---|---|---|
| Ankit Gupta, Shohei Koide | 2015 | Crystal structure of monobody Mb (S4) bound to Prdm14 in complex with Mtgr1 | http://www.rcsb.org/pdb/explore/explore.do?structureId=5ECJ | Publicly available at the RCSB Protein Data Bank (accession no. 5ECJ) |

| Nady N, Tomek Swigut, Joanna Wysocka | 2015 | ETO Family Protein Mtgr1 Mediates Prdm14 Functions in Stem Cell Maintenance and Primordial Germ Cell Formation | http://www.ncbi.nlm.nih.gov/geo/query/acc.cgi?acc=GSE74547 | Publicly available at the Gene Expression Omnibus (accession no. GSE74547) |
|---|---|---|---|---|

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
