## [Decision Letter]

Thank you for submitting your work entitled "ETO Family Protein Mtgr1 Mediates Prdm14 Functions in Stem Cell Maintenance and Primordial Germ Cell Formation" for peer review at *eLife*. Your submission has been favorably evaluated by Janet Rossant (Senior editor), a Reviewing editor, and three reviewers.

The reviewers have discussed the reviews with one another and the Reviewing editor has drafted this decision to help you prepare a revised submission.

Summary:

In this study, the Wysocka and Koide labs have examined how Prdm14, a PR-SET domain transcription factor regulates gene expression. Prdm14 has a critical role in pluripotency in ES cells and in the specification of primordial germ cells (PGCs). They have identified an ETO-family protein Mtgr1 as a co-repressor that binds tightly to the preSET/SET domain as shown by crystal structure of Prdm14-Mtgr1 complex. The authors mapped the domains of both proteins involved in direct interaction, demonstrated (via Chip-seq) that these two proteins tend to recognize the same genomic targets, probably through the DNA binding domain of Prdm14, and indicate that Mtgr1 is required for Prdm14 function in ESCs and PGC formation. Mtgr1 is known for its role in leukemias but had not been previously implied in the regulation of pluripotency and germ line formation. In fact, Mtgr1 null mice appear viable and fertile, suggesting some redundancy in vivo.

The authors also generated a collection of monobodies that either disrupt the interaction between Prdm14 and Mtgr1 or allow the co-immunoprecipitation of the two proteins. Using a monobody that does not inhibit Prdm14-Mtgr1 interaction and a tethered protein construct, the authors obtain crystals of Prdm14-Mtgr1-Mb(S4). The structure was used to analyze the effects of interface mutants.

Overall, the findings are novel and provide information, at least in part, on how Prdm14 functions to repress target genes. Previous studies had indicated Prdm14 interaction with PRC2 and TET1/2, which could not be confirmed in this study. Prdm14 can also activate some genes, but how this occurs remains to be determined in the future.

The authors apply a wide range of approaches, ranging from ESC studies (with knockouts) to structures, monobody generation, and genomics. Indeed, the monobody approach could represent a novel tool to study pluripotency and the regulation of gene expression, and open up mechanisms to target Prdm14 in cancers. With appropriate revisions, this paper is highly suitable for publication in *eLife*.

Essential revisions:

1) The conclusion from Figure 2 is that some Mtgr1 bound sites are Prdm14 independent. However, that may not be completely clear based on just over-expression experiments. The authors should acknowledge this (editing the manuscript to be more clear on this point) or add new experiments such as showing Mtgr1 ChIP at "Prdm14-independent" sites in a Prdm14 KO ESC line, or upon over-expression of mutant Prdm14 constructs.

Further, it would be informative to know if there are Prdm14 binding sites without Mtgr1. This might indicate that Prdm14 could recruit other interactors to targets, especially to putative targets that might be activated by Prdm14. The authors are encouraged to add new data to clarify these points, or alter the text to be more clear on what is formally shown by the existing data.

2) The RNA-seq data obtained from ESCs expressing mutant Prdm14 proteins (Figure 6) should be shown by PCA analysis and differential gene expression/heatmaps as in Figure 3, plus some correlation values/GO-terms, to better reveal the similarity between the samples. Also, it seems only one mutant's RNA-seq data are shown? Data from at least an n=2 is needed.

3) Loss of Prdm14 and Mtgr1 in ES cells led to the upregulation of target genes. Careful comparison of the RNA-Seq data might show if there are any differences between them. The authors are encouraged to perform such analyses and clarify which targets are in common and which are distinct.

4) The study shows that Mtgr1 (like Prdm14) is required for PGC specification. This suggests that Prdm14-Mtgr1 plays a similar role to that in ES cells. While this is a reasonable assumption, it does not discount additional or different targets (repressed and activated) in PGCs. it remains unclear whether Mtgr1 deletion does specifically affect germline differentiation or would lead to a general block in differentiation. Thus, the question remains whether the critical function of Mtgr1 is limited to the germline. The analysis of the self-renewing ESC state in the absence of Mtgr1 is rather limited (only RNA-seq and morphology). Expansion of these analyses to clarify whether Mtgr1 blocks cell differentiation more generally would be welcomed.

5) Published data on Mtgr1 knockout does not support the proposed role for this gene since the mutant mice are viable and fertile. This remains a major issue to be resolved in light of this study. While other ETO proteins might compensate for Mtgr1 in the germline in vivo, this would also be expected in the in vitro studies on PGCLCs. Some discussion of this conundrum is warranted.

6) Prdm14 and Mtgr1 should be shown to interact in wt ESCs that are not over-expressing any tagged proteins.

7) There is quite high percentage (>10%) of the protein residues located outside of favorable Ramachandran plot, which is unusual. The authors need to check their structure carefully, and/or give an explanation for such large number of residues in non-favorable conformations. For example, what were the values for *R_merge_* or *R_sym_* (in addition to *R_pim_*)?

8) A covalently linked construct of Prdm14-linker-Mtgr1 was used for crystallization, to overcome the aggregation problem. There could be potential issues with this approach by artificially forcing the two domains together in a certain way. To alleviate these concerns, a specific protease site could be introduced into the linker. The authors could probe whether, after cleavage, the two proteins still associate? Can the same crystal be obtained after adding Mb(S4)? I note that, based on the structure information provided (Figure 5), the construct could also be made in the order of Mtgr1-linker-Prdm14. Does this give the same crystal and/or structure? The reviewers find it important to provide data demonstrating that the specific linked construct used did not impact the structure presented.

9) The work on monobodies is of general interest. Here the work refers to the use of human PRDM14. Is there evidence to show that hPRDM14 can substitute for mouse Prdm14 in ES cells or PGCs? I note in the crystal structure studies that mouse Prdm14 SET domain is similar to human PRDM12 SET domain.

---

## [Author Response]

*Essential revisions: 1) The conclusion from Figure 2 is that some Mtgr1 bound sites are Prdm14 independent. However, that may not be completely clear based on just over-expression experiments. The authors should acknowledge this (editing the manuscript to be more clear on this point) or add new experiments such as showing Mtgr1 ChIP at "Prdm14-independent" sites in a Prdm14 KO ESC line, or upon over-expression of mutant Prdm14 constructs.*

In the original manuscript we uncovered two types of Mtgr1 genomic sites: first class contained Prdm14 sequence recognition motif, whereas the second class did not. Based on these observations we referred to the first class as the 'Prdm14-dependent' and to the second class as 'Prdm14-independent' sites. We further noted that the first class sites showed robust Prdm14 binding and gain of Mtgr1 ChIP-seq signal in the Prdm14 overexpressing cell line. In contrast, the second class showed low/no Prdm14 binding and diminished occupancy of Mtgr1 upon Prdm14 overexpression. However, we agree with the reviewers that our results did not show directly that Prdm14 is not required for Mtgr1 binding at the second class of sites, and in retrospect, we should have referred to these sites as 'Prdm14 motif-lacking' sites rather than 'Prdm14-independent'. This is now rectified in the revised manuscript.

To address the reviewers' comment further, we have now performed Mtgr1 ChIP-seq from Prdm14^-/-^ ESCs, and generated average signal profiles at Prdm14 motif-containing and Prdm14 motif-lacking sites across all our ChIP-seq datasets (shown in the revised Figure 2—figure supplement 2). The results can be summarized as follows: (i) in wt ESCs, the average Mtgr1 enrichment is similar at Prdm14 motif-containing vs. lacking sites (compare black profiles in left and right panels; note that the scale on the Y axis is the same), (ii) at Prdm14 motif-containing sites, Mtgr1 binding is increased in FH- Prdm14 OE cells and diminished (but not entirely abrogated) in Prdm14^-/-^ cells, left panel, (iii) at Prdm14 motif-lacking sites, Mtgr1 binding is diminished by FH-Prdm14 overexpression, but it is also moderately affected in Prdm14^-/-^ cells (right panel). Altogether, these results are consistent with the Mtgr1 binding to chromatin being sensitive to the Prdm14 dosage (either loss or gain) at the Prdm14-motif containing sites, which are the main focus of our study. However, these results also demonstrate that even in the absence of Prdm14, some Mtgr1 binding remains at the motif-containing sites, suggesting partial redundancies in the recruitment mechanisms. Finally, although quite likely a modest negative effect of Prdm14 deletion on Mtgr1 binding at motive-lacking sites can be attributed to indirect effects (given that these sites show low/no Prdm14 occupancy), we nonetheless concur that it is best to refrain from calling these sites Prdm14-independent.

We have now clarified some of these nuances in the text (subsection “Prdm14 and Mtgr1 co-occupy genomic targets”) and also included an additional figure showing the top five sequence motifs recovered at the Prdm14 motif-lacking Prdm14 sites (Figure 2—figure supplement 2). Of note, the top motif recovered at these sites represents a consensus motif for helix-loop-helix transcription factors, suggesting that a TF from this family may be mediating Mtgr1 recruitment to the Prdm14 motif-lacking sites.

*Further, it would be informative to know if there are Prdm14 binding sites without Mtgr1. This might indicate that Prdm14 could recruit other interactors to targets, especially to putative targets that might be activated by Prdm14. The authors are encouraged to add new data to clarify these points, or alter the text to be more clear on what is formally shown by the existing data.*

We have not been able to detect a class of Prdm14 sites devoid of Mtgr1 binding. This point is illustrated in Figure 2, showing Mtgr1 vs. Prdm14 ChIP-seq enrichments at all genomic sites bound by either Mtgr1 or Prdm14 in FH-Prdm14 overexpressing ESCs. If a class of sites bound strongly by Prdm14, but not Mtgr1 existed, than it would show up as a population of sites that have high signal at the X axis and low signal at the Y axis. However, we do not detect such population. As a side note with respect to the previous comment: given that this analysis was out of necessity (as we do not have a ChIP-compatible Prdm14 antibody) done in FH-Prdm14 OE cells, in which Mtgr1 binding at Prdm14 motif-lacking sites is highly diminished, the population of sites that have low value on the X axis and high value on the Y axis (e.g. bound by Mtgr1 and not Prdm14) is also relatively small. Consequently, Mtgr1 and Prdm14 binding patterns in FH-Prdm14 cells are highly correlated (R correlation coefficient ~0.9).

*2) The RNA-seq data obtained from ESCs expressing mutant Prdm14 proteins (Figure 6) should be shown by PCA analysis and differential gene expression/heatmaps as in Figure 3/C, plus some correlation values/GO-terms, to better reveal the similarity between the samples. Also, it seems only one mutant's RNA-seq data are shown? Data from at least an n=2 is needed.*

First, we want to clarify that data from two distinct cell lines reconstituted with Prdm14 E294K mutants were used in our analyses. Following the reviewers' suggestions, we have now performed the principal component analysis, differential gene expression, and clustering based on distances among our samples, including the two Prdm14^-/-^ cell lines reconstituted with the mutant Prdm14 protein. These data are shown in Figure 6—figure supplement 1. Altogether, these results support the notion that Prdm14 E294K lines show hypomorphic expression profile, which falls in between the Prdm14^-/-^ ESCs and those reconstituted with wt Prdm14. Nonetheless, cells reconstituted with Prdm14 E294K mutant cluster together with Prdm14^-/-^ and Mtgr1^-/-^ cells and away from cells rescued with wt Prdm14 (Figure 6—figure supplement 1). When considering PCA plot in Figure 6—figure supplement 1, please note the following: (i) cells reconstituted with wt Prdm14 have a gain-of-function phenotype, which results in tighter repression of differentiation genes and makes their gene expression profile look more like that of ESCs grown under 2i+LIF, even when they are maintained under serum, (ii) wt and mutant reconstituted cell lines show comparable Prdm14 overexpression levels.

*3) Loss of Prdm14 and Mtgr1 in ES cells led to the upregulation of target genes. Careful comparison of the RNA-Seq data might show if there are any differences between them. The authors are encouraged to perform such analyses and clarify which targets are in common and which are distinct.*

We have performed careful analysis of all genes that are upregulated at least twofold upon knockout of either Prdm14 or Mtgr1 (datasets from multiple clonal knockout lines have been considered for each), and we color coded them based on whether they are statistically significantly affected in one or both set of knockout samples (Figure 3—figure supplement 4 in the revised manuscript). The majority of genes upregulated in either knockout is upregulated in both (purple dots). However, there is also a subset of genes that show significant upregulation only in Mtgr1 null cells (blue dots) or only in Prdm14 null cells (red dots). To our knowledge, these differential genes do not appear to be associated with a coherent biological function or phenotype. Nonetheless, to facilitate further follow up work we have incorporated gene names for these uniquely upregulated transcripts into the figure.

*4) The study shows that Mtgr1 (like Prdm14) is required for PGC specification. This suggests that Prdm14-Mtgr1 plays a similar role to that in ES cells. While this is a reasonable assumption, it does not discount additional or different targets (repressed and activated) in PGCs. it remains unclear whether Mtgr1 deletion does specifically affect germline differentiation or would lead to a general block in differentiation. Thus, the question remains whether the critical function of Mtgr1 is limited to the germline. The analysis of the self-renewing ESC state in the absence of Mtgr1 is rather limited (only RNA-seq and morphology). Expansion of these analyses to clarify whether Mtgr1 blocks cell differentiation more generally would be welcomed.*

We thank the reviewers for raising this important point. To test whether loss of Mtgr1 results in general block of cell differentiation, we induced embryoid body (EB) formation from wild type and Mtgr1^-/-^ mESCs and analyzed expression of differentiation markers by RT- qPCR at day 4 and day 8 after induction. We detected comparable expression of markers representing all three germ layers in wild type and Mtgr1^-/-^ EBs (these results are shown in Figure 3—figure supplement 5). Although we cannot exclude a possibility that Mtgr1^-/-^ cells show differentiation defects in specific somatic lineages, our data are inconsistent with a general block in differentiation.

*5) Published data on Mtgr1 knockout does not support the proposed role for this gene since the mutant mice are viable and fertile. This remains a major issue to be resolved in light of this study. While other ETO proteins might compensate for Mtgr1 in the germline in vivo, this would also be expected in the in vitro studies on PGCLCs. Some discussion of this conundrum is warranted.*

We agree that this is a conundrum, as based on our results one would expect a germline phenotype in Mtgr1 knockout mouse, unless a stronger selection for the compensation by other ETO proteins occurs in vivo than in the vitro derived cells. Nonetheless, there are also caveats associated with the described targeting strategy in published Mtgr1 knockout mice, namely the first six exons encoding amino acids 1-316 (spanning the Prdm14 interaction region as well as NHR2 domain involved in dimerization and association with mSin3/HDAC complex), are still preserved (Amann et al., 2005; Rossetti et al., 2004). Thus, this mutant strain is likely hypomorphic. Furthermore, through a symposium presentation earlier this year we became aware of another Mtgr1 knockout study using a different targeting strategy, in which homozygous null mice showed very strong defects in both male and female germline. However, given that this work is as of yet unpublished and we are not familiar with all experimental details, unfortunately we cannot comment on these distinct results in our manuscript. We therefore hope that the reviewers can understand that we keep the discussion of the apparent discrepancy fairly succinct (from main text Discussion section):

“It is therefore surprising that Mtgr1 mouse knockout strain has been reported as viable and fertile (Amann et al., 2005). However, it remains unclear whether this strain represents a true loss-of-function, because in the mouse targeting strategy the first six exons are preserved (encoding amino acids 1-316 which span the Prdm14 interaction region and NHR2 domain involved in dimerization and association with mSin3/HDAC complex) (Rossetti et al., 2004). In light of our findings and aforementioned caveats, we suggest that the in vivo function of Mtgr1 in the germline should be revisited. It is also possible, however, that requirement for Mtgr1 in germ cell development in vivo can be fully or partially compensated by the other ETO proteins.”

*6) Prdm14 and Mtgr1 should be shown to interact in wt ESCs that are not over-expressing any tagged proteins.*

While IP experiments with endogenous proteins were initially hampered by the lack of suitable antibodies recognizing mouse Prdm14, we were now able to perform these analyses by taking advantage of our monobodies, which we used to precipitate endogenous Prdm14 from wild type ESCs. Immunoblot analysis with α-Mtgr1 antibody showed that Mb(S4) monobody, which does not disrupt Prdm14-Mtgr1 interaction, but not Mb(S14) monobody, which binds competitively, recovers endogenous Mtgr1 (and is able to deplete most of it from the extract). These data are now part of Figure 4—figure supplement 2. In addition, we now performed Prdm14 Mb(S4)-precipitation/mass spec analysis from wt ESCs, which readily detected Mtgr1-originating peptides (Figure 4—figure supplement 2).

*7) There is quite high percentage (>10%) of the protein residues located outside of favorable Ramachandran plot, which is unusual. The authors need to check their structure carefully, and/or give an explanation for such large number of residues in non-favorable conformations. For example, what were the values for* R_merge_
*or* R_sym_
*(in addition to* R_pim_*)?*

We thank the reviewer(s) for suggesting the possibility of improving the structure. We have now reexamined the structure and indeed we have been able to improve its geometries. The new structure has 95.8% favorable, 4.1% allowed and 0.1% outliers, which we consider as acceptable statistics. In addition, the newly refined structure has improved *R/R_free_* values of 0.189/0.250. The improved structure no longer has statistically unusual features.

*8) A covalently linked construct of Prdm14-linker-Mtgr1 was used for crystallization, to overcome the aggregation problem. There could be potential issues with this approach by artificially forcing the two domains together in a certain way. To alleviate these concerns, a specific protease site could be introduced into the linker. The authors could probe whether, after cleavage, the two proteins still associate? Can the same crystal be obtained after adding Mb(S4)? I note that, based on the structure information provided (Figure 5), the construct could also be made in the order of Mtgr1-linker-Prdm14. Does this give the same crystal and/or structure? The reviewers find it important to provide data demonstrating that the specific linked construct used did not impact the structure presented.*

We appreciate the importance of explicitly supporting the authenticity of the structure determined with the use of a flexible linker, although connecting an interacting pair of proteins with a flexible linker is a common approach to stabilize the complex while retaining its biological function. This linker strategy is well accepted among crystallographers (Ernst et al., 2014; Kobe et al., 2015; Reddy Chichili et al., 2013; Zhou et al., 2015). The linker we used in this work is similar to those used for the single-chain Fv form of antibodies in which the light and heavy chain domains are linked. There are thousands of single-chain Fv molecules that are fully functional and most likely retain the native conformation. Whereas the approach suggested by the reviewers is intriguing, the difficulties in handling the Mtgr1 protein at high concentrations and our complete failure in crystallizing unlinked constructs even with the monobody strongly suggest that it would be a long-term project in itself. Therefore, we have taken different approaches using NMR spectroscopy and affinity measurements and produced additional data that strongly support the authenticity of the structure.

First, we compared the _1_H-_15_N HSQC spectrum of _15_N-Prdm14 in complex with unlabeled Mtgr1 against that of _15_N-labeled Prdm14-linker-Mtgr1 fusion. Because the amide NMR chemical shifts are exquisitely sensitive to the local conformation and each amino acid residue (except for proline) gives rise to an HSQC cross peak, a _1_H-_15_N HSQC spectrum gives a comprehensive fingerprint of an _15_N-labeled protein. Most of the cross peaks in the HSQC spectrum of _15_N-Prdm14 in complex with unlabeled Mtgr1 (shown in red in Figure 5—figure supplement 1; where we observe signals only from _15_N-Prdm14) overlapped with those in the HSQC spectrum of _15_N-labeled Prdm14-linker-Mtgr1 (shown in black; where we observe signals from both Prdm14 and Mtgr1). The high levels of coincided peaks in the two spectra strongly suggest that the Prdm14 protein takes on nearly identical conformation in the unlinked complex and in the fusion protein (Figure 5—figure supplement 1). Please also note that the spectrum for _15_N-Prdm14 in complex with unlabeled Mtgr1 was of poorer quality because we needed to keep its concentration low to prevent aggregation.

Second, we determined the affinity of Mb(S4) to the Prdm14-linker-Mtgr1 fusion and to the Prdm14/Mtgr1 complex. The dissociation constants of Mb(S4) to these two samples were identical (Figure 5—figure supplement 1), indicating that the epitope for Mb(S4) on Prdm14 is not distorted by the introduction of the linkage between Prdm14 and Mtgr1.

In addition to the new data described above, we would like to note the following points already in the original manuscript that support the crystal structure:

i) The linked molecule maintained the 1:1 heterodimer architecture of the separate Prdm14-Mtgr1 complex in solution and in the crystal. Therefore, the linkage did not introduce gross changes to the complex, such as oligomerization due to 3D domain swapping (reviewed in (Liu and Eisenberg, 2002)). The size-exclusion chromatography data are now included as Figure 5—figure supplement 1.

ii) The linker used in our work is sufficiently long to allow the two domains to move freely. The distance between the beginning and end of the linker in the observed structure is ~18 Å, whereas the polypeptide connecting these points include eight residues that can span more than 24 Å even using a conserved (i.e. short) distance estimate of 3 Å per residue. Accordingly, the linker makes a U-turn in the structure and it cannot impose steric strain ("pull") between the two domains. Indeed, several residues within the linker are undetected, suggesting that they are flexible and not under tension. In retrospect, the dotted straight lines in original Figure 5 may have given a false sense of tension or distortion in the structure. We have replaced the line with a curve that more accurately reflects the actual length of the linker.

iii) The reviewer suggested, "*To alleviate these concerns, a specific protease site could be introduced into the linker. The authors could probe whether, after cleavage, the two proteins still associate?"* We have shown that free Prdm14 and Mtgr1 associate with high affinity (Figure 1), which in our opinion has already answered this question.

iv) The mutation data (Figure 6) validate the authenticity of the observed interface including a specific ionic interaction in the full-length Prdm14 and Mtgr1 proteins, and show that structure-based mutations affect key biological functions of Prdm14.

Altogether, we hope that this extensive series of results convinces reviewers that the linker strategy used here introduced little if any perturbation to the conformation of the Prdm14/Mtgr1 complex. In order to clarify this important point, we have added a paragraph that summarizes the discussion above (subsection “Generation of renewable monobody reagents to study and inhibit Prdm14-Mtgr1 complex”).

*9) The work on monobodies is of general interest. Here the work refers to the use of human PRDM14. Is there evidence to show that hPRDM14 can substitute for mouse Prdm14 in ES cells or PGCs? I note in the crystal structure studies that mouse Prdm14 SET domain is similar to human PRDM12 SET domain.*

First, we would like to point out that our monobodies recognize mouse and human Prdm14 with similar affinities (Figure 4). Regarding the second part of the comment: a question whether hPRDM14 can complement defects observed upon loss of the mouse protein has, to our knowledge, never been addressed. Therefore, to answer reviewer's point, we cloned hPRDM14 cDNA, and introduced it to mouse Prdm14_-/-_ ESCs using a piggyBac system. We subsequently compared tagged Prdm14 levels (by immunoblotting), cell morphology and expression of select Prdm14-bound genes (by RT- qPCR) in Prdm14_-/-_ ESCs reconstituted with either mouse or human Prdm14 (shown in Figure 4—figure supplement 1). These analyses revealed that at least with respect to ESC functions, the hPRDM14 could substitute for the mouse protein (see subsection “Generation of renewable monobody reagents to study and inhibit Prdm14-Mtgr1 complex”).